# BENCHMARKING ECG FMS: A REALITY CHECK ACROSS CLINICAL TASKS

**M A Al-Masud, Juan Miguel Lopez Alcaraz & Nils Strodthoff**[*]
AI4Health Division, Carl von Ossietzky Universität Oldenburg
Oldenburg, Lower Saxony, Germany
{m.a.al-masud,juan.lopez.alcaraz,nils.strodthoff}@uol.de

## ABSTRACT

The 12-lead electrocardiogram (ECG) is a long-standing diagnostic tool. Yet machine learning for ECG interpretation remains fragmented, often limited to narrow tasks or datasets. FMs promise broader adaptability, but fundamental questions remain: Which architectures generalize best? How do models scale with limited labels? What explains performance differences across model families? We benchmarked eight ECG FMs on 26 clinically relevant tasks using 12 public datasets comprising 1,650 regression and classification targets. Models were evaluated under fine-tuning and frozen settings, with scaling analyses across dataset sizes. Results show heterogeneous performance across domains: in adult ECG interpretation, three FMs consistently outperformed strong supervised baselines. In contrast, ECG-CPC, a compact structured state-space model, dominated 5 of 7 task categories, demonstrating that architecture matters more than scale. FMs improved label efficiency 3.3-9× over supervised baselines, though scaling behaviors varied across architectures. Representation analysis reveals that models with similar performance learn markedly different internal structures, suggesting multiple viable paths to effective ECG representation. Overall, while FMs show promise for adult ECG analysis, substantial gaps remain in cardiac structure, outcome prediction, and patient characterization. ECG-CPC's strong performance despite being orders of magnitude smaller challenges the assumption that FM quality requires massive scale, highlighting architectural inductive biases as an untapped opportunity.

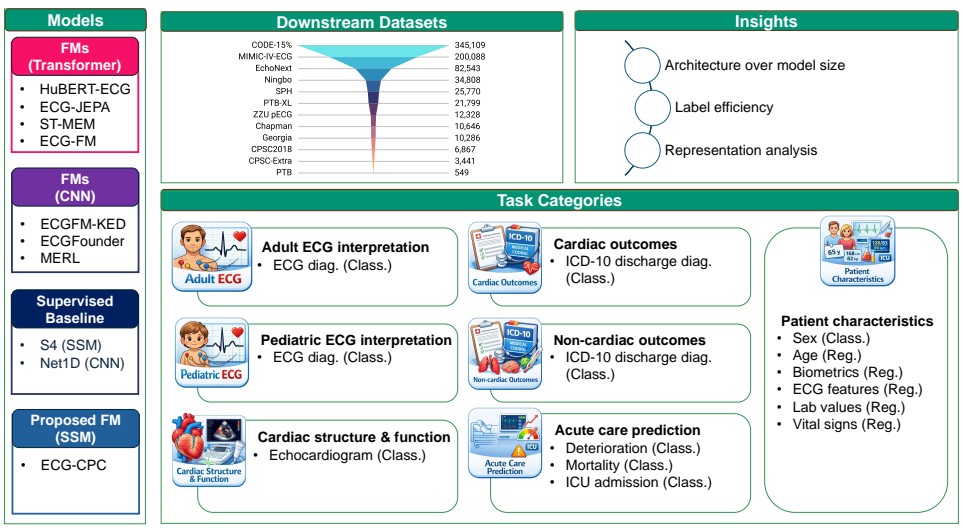

Figure 1: Overview of the benchmarking pipeline for ECG FMs.

[*]AI4Health Division — https://uol.de/en/ai4health

# 1 INTRODUCTION

**Clinical relevance** Electrocardiography (ECG) is a widely used non-invasive tool for assessing cardiac function and systemic physiology (Siontis et al., 2021). Accurate interpretation is essential for detecting myocardial infarctions (Strodthoff et al., 2020), evaluating cardiovascular risk (Bhatia & Dorian, 2018), and guiding clinical decisions (Rokos et al., 2010). ECG features also reflect systemic conditions such as electrolyte imbalances (Diercks et al., 2004), metabolic disorders (Wald, 2006), and physiological factors (Lopez Alcaraz et al., 2025), broadening its clinical utility. The growing availability of large-scale ECG datasets (Wagner et al., 2022; Gow et al., 2023; Ribeiro et al., 2021) has made machine learning increasingly pivotal for automated interpretation, enabling population-scale screening (Kalmady et al., 2024) and supporting clinical workflows (Graf et al., 2024).

**Promise of FMs for ECG** We follow the original definition from the Stanford Institute for Human-Centered Artificial Intelligence (Bommasani et al., 2022) and define foundational models (FMs) as "models trained on broad data (generally using self-supervision at scale) that can be adapted to a wide range of downstream tasks". FMs have transformed fields such as natural language processing (Myers et al., 2024) and computer vision (Awais et al., 2025), where large-scale pretraining produces robust and transferable representations across tasks and domains (Subramanian et al., 2023). FMs now also emerge in biomedical domains, including pathology (Chen et al., 2024), retinal imaging (Zhou et al., 2023), and biomedical time series, where ECG serves as a natural testbed given its ubiquity and clinical relevance. Clinical FMs offer three key advantages: (i) higher predictive performance than training from scratch, (ii) improved label efficiency via pretrained representations, and (iii) the utility of FMs as frozen feature extractors for downstream tasks. These benefits could enable stronger predictive models, support rare-disease studies on limited data, and guide the choice of FMs for specific applications.

**Research gap** Despite their promise, ECG FMs remain inadequately evaluated. Prior studies compare FMs against weak baselines (Li et al., 2025; Kim, 2024; Na et al., 2024; Liu et al., 2024), and existing benchmarks focus on narrow datasets or single task categories, preventing generalizable conclusions about model capabilities (Na et al., 2024; Liu et al., 2024). Critically, fundamental questions remain unanswered: Do larger FMs consistently outperform smaller, task-specific models? Which architectural choices from Transformers, CNNs, or state-space models yield the most transferable representations? How do different pretraining strategies affect label efficiency across clinical domains? Without systematic comparison under controlled conditions, the field cannot distinguish genuine advances from benchmark overfitting or favorable task selection.

**Contributions of this work** This work addresses three research questions central to ECG FM development: **Architecture:** Which backbone architectures generalize best across diverse ECG tasks?; **Efficiency:** How do FMs scale with labeled data compared to supervised baselines?; **Representation:** What explains performance differences between architecturally distinct models?

We present a comprehensive benchmark covering 8 FMs, 12 datasets, and 26 clinical tasks across classification and regression. Our key findings are: **(1) Architecture matters more than scale.** Structured state-space models (SSMs) outperform Transformer-based models on 5 of 7 task categories despite being much smaller. We introduce **ECG-CPC**, a lightweight SSM trained with minimal resources, which matches or exceeds all evaluated models, challenging the assumption that FM quality scales primarily with parameter count. **(2) FMs improve label efficiency 3.3-9×.** Scaling analysis shows strong models substantially reduce labeled data requirements compared to supervised baselines, especially for outcome prediction, though gains vary across architectures, guiding model selection under data constraints. **(3) Similar performance arises from divergent representations.** Centered kernel alignment (CKA) (Kornblith et al., 2019; Nguyen et al.) analysis reveals that models with comparable downstream performance learn markedly different internal feature structures, suggesting multiple viable paths to effective ECG representation and raising questions about whether task performance alone suffices for evaluating FMs.

# 2 BACKGROUND

**AI-enhanced ECG** Machine learning and deep learning now underpin automated ECG interpretation, improving arrhythmia detection, risk stratification, and clinical decision support. Models of

different architectural flavors, from CNNs to RNNs to Transformers, trained through supervised learning, have been shown to achieve strong performance. However, they require large, curated datasets and often generalize poorly across populations, devices, and settings (Ribeiro et al., 2020; Hannun et al., 2019; Hong et al., 2020). Benchmarking robust supervised baselines across multiple datasets remains crucial for assessing the benefits of representation learning and pretraining (Strodthoff et al., 2020; Nonaka & Seita, 2021; Hong et al., 2020).

**ECG FMs** Inspired by the success of FMs in language and vision, several approaches have been proposed for ECG. Architectures include CNNs (mainly ResNet variants), transformers (often with convolutional encoders), and structured state-space models. Pretraining methods vary from supervised and weakly supervised to contrastive and non-contrastive self-supervision. Pretraining datasets also differ widely, from Computing in Cardiology 2021 challenge subsets (Reyna et al., 2021b) to MIMIC-IV-ECG (Gow et al., 2023) to the large-scale HEEDB dataset (Koscova et al., 2024).

Table 1: Summary of the eight ECG FMs and two supervised baselines evaluated in this work, including their backbone architectures, training paradigms, pretraining datasets and sizes, and parameter sizes. Pretraining datasets: a: HEEDB; b: Chapman, Ningbo, CODE-15%; c: MIMIC-IV-ECG; d: CODE, PTB, CPSC2018, CPSC-Extra, PTB-XL, Georgia, Chapman, Ningbo, Hefei, SPH, MIMIC-IV-ECG; e: CPSC2018, CPSC-Extra, PTB-XL, Georgia, Ningbo, Chapman, MIMIC-IV-ECG; * indicates models that were trained in this work, all other models were taken from the literature.

| Name | Backbone | Pretraining | Pretraining Samples | Parameters |
|---|---|---|---|---|
| ECGFounder | CNN | Supervised | $10.7M^a$ | 33.8M |
| ECG-JEPA | Transformer | JEPA | $174k^b$ | 87.2M |
| ST-MEM | Transformer | Masked Autoencoder | $174k^b$ | 90.3M |
| MERL | CNN | Weak sup., contrastive | $800k^c$ | 4.6M |
| ECGFM-KED | CNN | Weak sup., contrastive | $800k^c$ | 9.7M |
| HuBERT-ECG | Transformer | MLM | $9.1M^d$ | 97.2M |
| ECG-FM | Transformer | MLM+contrastive | $1.5M^e$ | 93.9M |
| ECG-CPC* | SSM | CPC | $10.7M^a$ | 3.8M |
| Net1D* | CNN | - | - | 33.8M |
| S4* | SSM | - | - | 2.2M |

**Benchmarking FMs** The benchmarking of FMs has a long tradition in other fields such as computer vision (Goldblum et al., 2023) and NLP (Chang et al., 2024). In the medical domain, most efforts have so far focused on medical imaging (Neidlinger et al., 2025; Lee et al., 2025). Recently, benchmarking results for EEG FMs have been put forward (Xiong et al., 2025). However, no large-scale, comprehensive benchmark for ECG FMs exists.

## 3 METHODS

### 3.1 MODELS

Table 1 summarizes the investigated models, including backbones, pretraining methods, and datasets, and parameter counts. As custom pretraining is infeasible, only FMs with publicly available pretrained weights are included and integrated via wrapper modules into a common evaluation framework. We benchmark eight FMs and two supervised baseline models as described below. Minor discrepancies in parameter counts stem from adjusting the classification head to each dataset's label count, and for multimodal models we use only the signal encoder, focusing on ECG signals rather than full released models with text or other modalities; note that these encoders retain an implicit advantage from having been pretrained alongside additional modalities.

**FMs** We consider diverse architectures and pretraining strategies. ECGFounder (Li et al., 2025) uses a RegNet-inspired CNN pretrained on HEEDB with a supervised loss. ECG-JEPA (Kim, 2024) employs a transformer-based joint embedding predictive architecture (JEPA) (Assran et al., 2023). We use the multi-block variant. ST-MEM (Na et al., 2024) uses a ViT-1D transformer trained as a

masked autoencoder (He et al., 2022). MERL (Liu et al., 2024) applies contrastive text-signal alignment (Radford et al., 2021), with the ResNet18 variant evaluated. ECGFM-KED (Tian et al., 2024) uses a ResNet backbone and minimizes a contrastive loss between signals and ECG report text. HuBERT-ECG and ECG-FM (McKeen et al., 2025) are transformer-based, pretrained via masked language modeling, with ECG-FM additionally incorporating a sequence-level contrastive loss. Finally, we pretrain a structured state-space (SSM) model on HEEDB using contrastive predictive coding (CPC) (van den Oord et al., 2018; Mehari & Strodthoff, 2023) (see Supplementary Material).

**Supervised baselines** We evaluate two backbones trained from scratch: Net1D, from ECGFounder (Li et al., 2025), and S4 (Mehari & Strodthoff, 2023), a structured state-space model known for its ability to capture long-range dependencies (Gu et al., 2022). S4 has proven to be a strong baseline on PTB-XL and other datasets (Mehari & Strodthoff, 2023; Strodthoff et al., 2024).

## 3.2 DATASETS AND BENCHMARKING TASKS

We group the benchmark datasets into seven categories based on the qualitative nature of the underlying labels. See Table 2 in the Supplementary Material for details on sample sizes and task characteristics. **Adult/pediatric ECG interpretation:** These two categories, the most studied in the literature, involve predicting diagnostic ECG statements from cardiologists. Benchmarks include PTB (Bousseljot et al., 1995), Ningbo (Reyna et al., 2022), CPSC2018 and CPSC-Extra (Reyna et al., 2021a; 2022), Georgia (Reyna et al., 2021a; 2022), Chapman (Reyna et al., 2021a; 2022), SPH (Liu et al., 2022a;b), CODE-15% (Ribeiro et al., 2021), and PTB-XL (Wagner et al., 2022; 2020). A separate category includes ZZU pECG (Tan et al., 2025; Jian et al., 2025) for pediatric ECG interpretation. **Cardiac structure and function:** This category predicts outcomes from complementary modalities, here echocardiography. We use the EchoNext dataset (Elias & Finer, 2025), which is used to capture cardiac structure and function. **Cardiac and non-cardiac outcomes:** These categories cover the prediction of cardiac and non-cardiac discharge diagnoses from the first emergency department ECG (Strodthoff et al., 2024), distinguishing cardiac (ICD-10 chapter I) and non-cardiac diagnoses. **Acute care predictions:** ECG is a key modality for acute care decisions. We predict clinical deterioration, mortality at multiple time frames, and intensive care unit (ICU) admission (Alcaraz et al., 2025), training a joint model for cardiac, non-cardiac, and acute care outcomes to save computational resources. **Patient characteristics:** This category includes tasks where ECGs predict non-diagnostic patient characteristics, such as sex, age, biometrics, ECG features, laboratory values (Alcaraz & Strodthoff, 2025), and vital signs (Gow et al., 2023). The datasets span small specialized cohorts to large population studies, covering diverse classification and regression tasks. We train a single model for all tasks in this category using combined classification and regression losses.

## 3.3 METHODOLOGY

**Finetuning/linear evaluation** Pretrained encoders are augmented with a linear head matching each downstream task. All models use the standard 12 ECG leads without additional resampling or filtering. Training optimizes binary cross-entropy (classification) or MAE (regression) with AdamW (learning rate $1 \times 10^{-3}$, weight decay $1 \times 10^{-3}$), batch size 64, for 100 epochs, with model selection on the validation set via macro AUROC or MAE. To mitigate scale interference in multi-target regression, targets are z-normalized using training-set statistics. During fine-tuning, we apply layer-dependent learning rates: model backbones are split into two parts whose learning rates are reduced by factors of 100 and 10 relative to the prediction head (see Table 34 for ablation). Batch-normalization statistics are frozen for linear and frozen evaluation. For frozen evaluation, the linear head is replaced with a learnable query-attention head (Bardes et al., 2024), which uses a single trainable query vector to attend over encoder tokens and produce a pooled representation, providing a lightweight, parameter-efficient pooling mechanism for frozen embeddings. When supported by the architecture, we use 2.5-second ECG segments instead of full 10-second recordings, as longer windows substantially increase compute with minimal performance gain. At inference, predictions are averaged across four non-overlapping 2.5-second segments, improving performance over single-segment evaluation. These choices follow established discriminative ECG practice (Mehari & Strodthoff, 2023). In line with modern robustness recommendations, we do not fix a global random seed.

**Evaluation** For classification, we report macro-averaged AUROC as the primary measure of overall discriminative performance. For regression, we report mean absolute error (MAE) averaged across z-normalized predictions and targets. Per-label AUROC and MAE values are provided in the Supplementary Material. Statistical uncertainty is estimated via empirical bootstrapping on the test set ($n = 1,000$). Pairwise model comparisons are conducted by bootstrapping performance differences; differences whose 95% confidence interval excludes zero are deemed significant. Rankings are derived from these tests, with ties indicating no significant difference, thereby reflecting both relative performance and statistical uncertainty across task categories. We adopt macro AUROC as the primary metric because it is the standard threshold-free ranking measure of discriminative ability. For clinical deployment, threshold-dependent metrics (e.g., sensitivity and specificity at label-specific operating points) are also relevant; an exploratory consistency analysis of rankings under such metrics is provided in Appendix A.10. Moreover, recent theoretical and empirical evidence (McDermott et al., 2024) indicates that AUROC is generally more reliable than alternatives such as AUPRC, particularly under label imbalance. Zero-shot evaluation is excluded because most ECG foundation models in our benchmark lack semantic label supervision, making zero-shot predictions neither meaningful nor comparable across architectures. Instead, consistent with current best practice for medical time-series FMs and the definition of foundation models in Bommasani et al. (2022), we focus on linear probing, frozen-feature evaluation, and full fine-tuning, which more reliably assess representation quality and adaptability.

## 4 RESULTS

**Results overview** Full finetuning, frozen evaluation, and linear evaluation results are given in Tables 3, 4, and 5 (Supplementary Material). Tables indicate statistically significant differences compared to the respective best method in this task. Ranked lists for all tasks and models are provided in Table 6. We further summarize this by reporting median ranks across all tasks of a given category in Figure 2. Appendix A.4 presents comparative predictions for the 10 best-performing conditions per task, illustrating both prediction quality and label diversity.

### 4.1 SUPERVISED BASELINES

**Supervised baseline vs. literature** Our supervised baseline performs on par with or surpasses the literature results. The S4-based model achieves AUROCs of 0.941 (vs. 0.9417 (Mehari & Strodthoff, 2023)) on PTB-XL, 0.908 (vs. 0.843 (Strodthoff et al., 2024)) for cardiac discharge, 0.849 (vs. 0.764 (Strodthoff et al., 2024)) for non-cardiac discharge, 0.863 (vs. 0.752 (Alcaraz et al., 2025)) for clinical deterioration, 0.747 (vs. 0.746 (Alcaraz et al., 2025)) for ICU admission, and 0.874 (vs. 0.816 (Alcaraz et al., 2025)) for mortality. Improvements on MIMIC-based tasks (except for ICU) are attributed to the multi-task training objective.

**Impact of model architecture** We also compare the supervised S4 model with the convolutional Net1D backbone used in ECGFounder, excluding supervised transformers due to their poor performance. Table 3 reports numerical results, and Table 6 provides statistically significant rankings: S4 ranks first across tasks, while Net1D is typically four or more positions lower. Although not our primary focus, this backbone comparison reinforces prior findings that CNNs are suboptimal for physiological time series (Strodthoff et al., 2024).

### 4.2 FINETUNING

**Adult ECG interpretation:** Across 11 tasks on 9 datasets, ECGFounder, ECG-JEPA, and ECG-CPC are the top performers, often statistically surpassing the S4 baseline. ECG-FM ranks fourth overall, occasionally matching the leaders but underperforming on Georgia, Chapman, and PTB-XL, while MERL, ST-MEM, HuBERT-ECG, and ECGFM-KED generally fail to outperform S4. **Pediatric ECG interpretation:** ECG-JEPA leads, followed by ECGFounder, ST-MEM, MERL, ECG-CPC, and S4, despite the absence of pediatric pretraining data. **Cardiac structure & function:** ECG-CPC ranks first for echocardiography predictions, followed by ECGFounder, ECG-JEPA, ST-MEM, MERL, and S4. **Cardiac and non-cardiac outcomes:** ECG-CPC dominates, matching ECG-FM on non-cardiac and S4 on cardiac outcomes. ECGFounder performs relatively poorly, likely due to limited overlap between pretraining and target diagnostic labels. **Acute care predic-**

**tions:** ECG-CPC and ECG-FM perform best across three tasks, followed by ECGFounder, ECG-JEPA, and MERL, but none significantly exceed S4. **Patient characteristics:** ECG-CPC ranks first in 5 of 6 tasks, surpassing S4 in 3; MERL and ECG-FM generally match or slightly trail S4, whereas ECGFounder and ECG-JEPA typically fall below it.

## 4.3 FROZEN AND LINEAR EVALUATION

**Frozen vs. supervised** Model rankings under frozen evaluation largely mirror finetuning results, with some differences. ECGFounder and ECG-JEPA continue to perform strongly on adult ECG interpretation, matching the supervised baseline, while ECG-CPC ranks slightly lower, with a median rank of 3. ECG-JEPA still dominates pediatric ECG interpretation. In other categories, ECG-CPC continues to lead. Notably, ECGFounder and ECG-JEPA can serve as effective frozen feature extractors for adult ECG interpretation tasks, achieving supervised-level performance. In the remaining categories, this applies to selected tasks for ECG-CPC under frozen evaluation and for ST-MEM under linear evaluation.

**Finetuning vs. frozen/linear** Model rankings under finetuning largely mirror frozen/linear evaluation. Strong models like ECG-JEPA, ECGFounder, and especially ECG-CPC maintain strong performance across other evaluation modes. However, some models such as MERL, and ECG-FM rely more on finetuning to reach competitive rankings, Interestingly, ST-MEM shows a much better relative ranking under linear evaluation than finetuning. This highlights that finetuning and linear/frozen evaluation relate to largely similar, though not completely congruent, aspects of representational quality and should therefore both be considered.

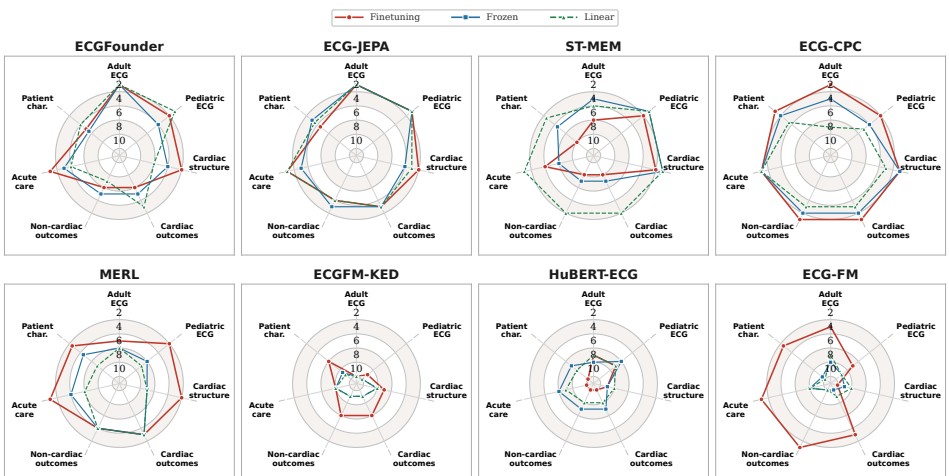

Figure 2: Radar plots summarizing model performance ranks (lower rank indicates statistically significantly better performance) for the eight FMs across the 7 investigated tasks. Our investigated ranking criteria accounts for confidence interval overlaps within each of the tasks and datasets. The plot is based on data from Table 7.

**Frozen vs. linear** Most methods are only slightly affected when using a linear instead of a non-linear prediction head. Deviations from this pattern are ST-MEM, which ranks among the best FMs under linear evaluation unlike in frozen evaluation, and ECG-CPC, which consistently underperforms with a linear head. The latter effect likely relates to pretraining: supervised or global contrastive objectives encourage discriminative pooled representations, unlike purely token-level pretraining. These results can be seen as an incentive to combine token- and sequence-level objectives, as done in ECG-FM and also commonly observed in computer vision (Caron et al., 2021). However, we advocate frozen evaluation as a less biased measure of FM representational quality than finetuning.

## 4.4 LABEL EFFICIENCY

**Label efficiency** Scaling curves for three FMs are shown in Figure 3 with scaling parameters listed in Table 30. All models stay below the supervised baseline (S4) throughout the entire range of

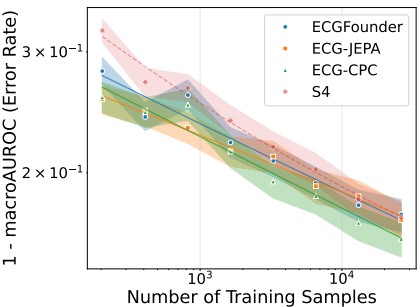

Figure 3: Scaling with dataset size on EchoNext across the best-performing FMs and supervised baseline.

**Setup** To isolate the effect of dataset size from task difficulty, we perform a controlled scaling experiment for the cardiac structure & function prediction on EchoNext (incorporating only labels with $>10,000$ counts). Training and validation subsets are subsampled in powers of 2 down to 1/128, using multi-label stratification (Wagner et al., 2022) on diagnostic labels, age bins, and sex. We finetune ECGFounder, ECG-JEPA, and ECG-CPC on these subsets, comparing pretrained models to training from scratch, with the S4 supervised baseline as reference. Performance is plotted as $1 - \text{macro AUROC}$, and scaling curves are fitted as $CN^{-\alpha} + L_0$, where $C$ is a constant, $N$ is the size of the training set, $\alpha$ the scaling exponent, and $L_0$ the residual error.

considered training set sizes. We use the parametric form of the fits to work out a label efficiency ratio $r = N^*/N$, i.e., the fraction of samples $N^*$ required for the pretrained model to reach the same performance as the supervised baseline for given $N$. For $N$ in the range of 250 to 1000, this yields label efficiency ratios between 0.30-0.62 for ECGFounder, 0.11-0.42 for ECG-JEPA and 0.21-0.40 for ECG-CPC, see Table 31 for details. This positions ECG-JEPA as the most label-efficient model, in particular in the very low sample size regime, closely followed by ECG-CPC. This establishes the label efficiency as a relevant benchmark parameter for FMs. Furthermore, the results show that ECG FMs fulfill the promise of an improved label efficiency not only in comparison to the respective model architecture trained from scratch but also against a strong supervised baseline, improving label efficiency by up to a factor of 9. Beyond absolute label efficiency, our scaling analysis distinguishes between data-efficiency slope and performance ceiling. ECG-JEPA learns quickly with few samples (steep slope) but plateaus at a lower accuracy than ECG-CPC, which improves more slowly yet reaches a higher ceiling. Thus, FMs differ in both how fast and how far they improve as data increases, suggesting that model choice should weigh slope versus ceiling depending on whether rapid low-label learning or maximal final performance is the priority. For the case of cardiac structure analysis, this establishes ECG-JEPA as superior model in the small sample regime ($< 10^3$ training samples) and ECG-CPC as superior model in the large sample regime, thereby providing an actionable recommendation for practitioners.

## 4.5 REPRESENTATION SIMILARITY ANALYSIS

We analyze intra-model CKA patterns for FMs in their original, unfinetuned form (Figure 4). ECG-CPC shows a progression from redundant early CNN layers to distinct S4 layers, with the final S4 block most specialized. ECGFounder's convolutional stem is dissimilar to higher layers, while mid-network layers (S0-S4) are highly similar, with specialization emerging only in the final layers. In ECG-JEPA, the embedding layer is distinct, but intermediate transformer blocks (Blk1-Blk10) are nearly identical, with only the final block specialized.

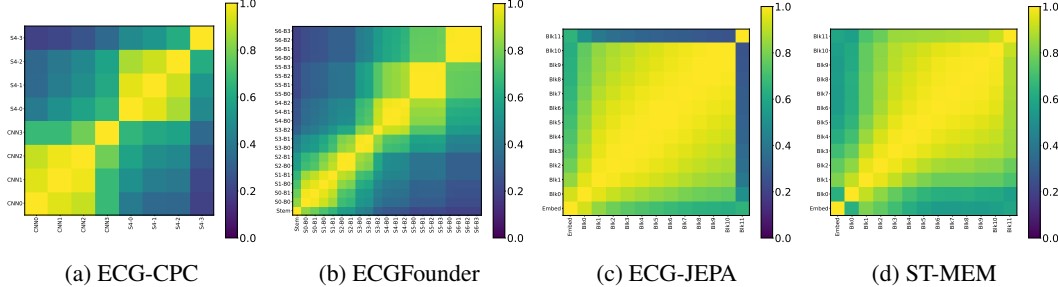

(a) ECG-CPC          (b) ECGFounder          (c) ECG-JEPA          (d) ST-MEM

Figure 4: Intra-model layer-wise representation similarity analysis. CKA heatmaps comparing all internal layers within each of the best performing FM on PTB-XL (all) dataset. Higher values (yellow) indicate similar representations between layers. CKA computed using Gaussian RBF kernel ($\sigma = 1.0$) on 2,500 samples per model.

ST-MEM exhibits a similar pattern, with a mostly homogeneous mid-section and specialized final layers, while the first transformer block stands out as dissimilar to neighboring blocks indicating a possible architectural bottleneck. Overall, CKA highlights that ECG-CPC exhibits the clearest and most structured evolution of representations, while other architectures show mid-layer redundancy or homogeneous transformer blocks. For a more detailed discussion including inter-model representation comparisons across depths, see Section A.9 in the supplementary material.

## 4.6 MODEL EFFICIENCY

For practical deployment, we benchmark the investigated models across GFLOP counts, GPU memory usage, and inference efficiency. Although SSM-based models are not as computationally efficient as CNNs in terms of inference efficiency, SMM (built on CPC) remains several times more efficient than transformer-based approaches across FLOPs, memory, and inference speed. More importantly, the efficiency gap between CNNs and SSMs becomes less critical when considering model quality: SMM delivers markedly stronger predictive performance than both CNNs and transformers, offering a balanced trade-off between computational cost and superior representation quality-making it an attractive choice for practical, real-world ECG deployment. The full results are given in Table 32 in the supplementary material.

## 5 DISCUSSION

**Details matter** Reliable downstream performance depends on careful tuning. Layer-dependent learning rates consistently improve results, and some models (e.g., HuBERT-ECG, ECG-FM) fail to train at all without them. As detailed in Table 34, this benefit is strongly architecture-dependent, with transformer and SSM-based models showing marked improvements with layer-dependent learning rates. Among CNN-based models, ECGFounder remains invariant, ECGFM-KED degrades, while MERL shows modest improvements. Adjustable input size (where possible) also matters: using 2.5-second crops with test-time averaging outperforms the full 10 seconds inputs.

**Comparable assessment** ECGFounder, ECG-JEPA, and ECG-CPC are strong on adult ECG interpretation, sometimes surpassing the supervised baseline. Pediatric ECG is dominated by ECG-JEPA. Other categories are led by ECG-CPC, followed by ECG-FM, while ECGFounder and ECG-JEPA are weaker here. Several models fall clearly below the supervised baseline, showing that self-supervised pretraining does not necessarily yield effective downstream performance. No single model consistently excels across all tasks and evaluation modes, though ECG-CPC comes closest to this goal with finetuning. This underscores that FM selection for downstream tasks should be informed by benchmarking results and similarity of the downstream task to the task categories considered here.

**Pretraining strategies** This benchmark cannot definitively determine the optimal pretraining method, as downstream performance reflects the interaction of pretraining strategy, dataset, and model architecture (Table 1). Superior supervised performance may favor SSMs over CNNs, while transformers align naturally with masking used in self-supervised methods. Larger datasets can help (ECGFounder, ECG-CPC) but are not sufficient (e.g., HuBERT-ECG), and large-scale models do not automatically generalize. ECG-CPC's strong performance outside ECG interpretation suggests self-supervised pretraining can be advantageous, though its unidirectional backbone limits supervised performance relative to bidirectional models. These results underscore the need for like-for-like pretraining comparisons on a common dataset (e.g., HEEDB) with standardized architectures. Fully disentangling dataset, architecture, and training strategy remains future work; our benchmark reveals practical differences, but controlled unified-dataset ablations are needed to explain them.

**Model architecture** In our benchmark, we provide the first systematic evidence that compact SSMs outperform much larger Transformers and comparable CNNs on time-series tasks despite having far fewer parameters, raising the theoretical question: what properties of SSM representations enable such efficient learning at minimal capacity? These models carry strong inductive biases, stable long-range memory, smooth spectral filtering, and globally parameterized convolutions, that align exceptionally well with the structure of ECG signals, enabling efficient learning even at small scale. This empirical insight, provides grounding that can catalyze new theoretical work.

**S4 among SSMs** Although S4 is an earlier state-space model, it remains a strong and widely used SSM baseline for continuous clinical time series due to its stable memory, spectral filtering, and computational efficiency, all of which align well with physiological signal structure. Prior ECG/EEG studies (Mehari & Strodthoff, 2023; Wang & Strodthoff, 2025) and recent SSM surveys (Somvanshi et al., 2025; Patro & Agneeswaran, 2025), consistent with our internal experiments shown in Appendix A.8, show that newer variants such as Mamba often excel in discrete domains but do not consistently outperform S4 on continuous medical signals, supporting its inclusion as a competitive and representative SSM baseline.

**Model complexity** The evaluated models differ markedly in size, computational efficiency, and predictive performance. Parameter counts increase from SSMs (smallest) to CNNs to transformers (largest). In terms of practical efficiency, considering GFLOPs, GPU memory, and inference latency, CNNs are fastest, followed by SSMs, with transformers being the least efficient. Importantly, SSM-based models, despite moderate computational cost, achieve the highest predictive performance, outperforming both transformers and CNNs, which show comparable accuracy.

**Model insights** Frozen and linear evaluation reveal the nature of learned representations and the implicit knowledge captured by different FMs. Insights from probing, as used here, remain coarse, but methods from explainable AI such as analyzing representation structures and their alignment across layers and models (Vielhaben et al., 2025) could provide more detailed insights into the knowledge acquired by these models.

**Limitations** This work has several limitations. First, the proposed tasks include only in-distribution tests; out-of-distribution evaluation would require additional datasets with compatible labels. While diagnostic label mismatches make this challenging, it is feasible for demographic data. Second, multi-task models generally improve classification performance, but for regression we applied z-normalization to mitigate issues. Still, the multi-task model used for computational efficiency may underperform single-task models on certain tasks. Third, the consistent benefit of layer-dependent learning rates during finetuning indicates potential for improved finetuning strategies. Fourth, since models are pretrained on different datasets, isolating which factors drive performance differences is difficult. Retraining on a unified dataset would enable cleaner comparisons but is computationally prohibitive. Following standard practice in vision and NLP, we prioritize practical utility by evaluating models as released; controlled unified-dataset comparisons remain important future work.

## 6    CONCLUSION

In this work, we present a comprehensive benchmark of ECG foundation models (FMs) across seven task categories. Performance varied across domains: ECGFounder, ECG-JEPA, and ECG-CPC excelled in adult ECG interpretation, while ECG-CPC outperformed others in categories where many FMs failed to match the supervised baseline. Overall, selected ECG FMs show promise, outperforming strong supervised baselines during finetuning and achieving comparable performance when used as frozen feature extractors. ECG-CPC, introduced here, is a small-scale model based on a structured state-space backbone, trained on a single NVIDIA L40 GPU for three weeks. Its strong benchmark performance highlights opportunities for further improving ECG FMs. Code and model weights are provided in the supplementary material and will be publicly released. LLMs were used only for language refinement. Our ECG-CPC framework and weights are available at https://github.com/AI4HealthUOL/ecg-fm-benchmarking.

ACKNOWLEDGMENTS

This work received funding by the German Research Foundation (project 553038473).

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

# A  APPENDIX

## A.1  BENCHMARKING TASKS

Table 2 provides a detailed overview of benchmarking tasks and the related datasets.

Table 2: This table provides an overview of the datasets and prediction tasks included in this study, covering routine cardiac diagnostics, clinical outcomes, and patient metadata, along with their sample sizes, patient counts, and label structures. For all tasks, we report effective sample sizes rather than the total number of available ECGs. In classification tasks, the effective sample size corresponds to the number of ECGs and patients with at least one positive label, since samples containing only negative labels do not contribute information about the presence of a condition. In regression tasks, it corresponds to the number of ECGs and patients with at least one valid numeric target value, as missing targets cannot be used for training or evaluation. Consequently, effective sample sizes may be substantially smaller than the total dataset size. We adopt this definition to provide a realistic estimate of usable data per task, ensure comparability across tasks, and avoid overstating the available training signal. [†]: evaluation only

| Task | Dataset | Type | Samples | Patients | Outputs |
|------|---------|------|---------|----------|---------|
| **Adult ECG interpretation** | | | | | |
| ECG interpretation | PTB | Multi-label | 549 | 290 | 22 |
| ECG interpretation | Ningbo | Multi-label | 34,808 | unknown | 68 |
| ECG interpretation | CPSC2018 | Multi-label | 6,867 | unknown | 9 |
| ECG interpretation | CPSC-Extra | Multi-label | 3,441 | unknown | 33 |
| ECG interpretation | Georgia | Multi-label | 10,286 | unknown | 50 |
| ECG interpretation | Chapman | Multi-label | 10,646 | 10,646 | 42 |
| ECG interpretation | Chapman (rhythm)[†] | Multi-label | 10,646 | 10,646 | 9 |
| ECG interpretation | SPH | Multi-label | 25,770 | 24,666 | 35 |
| ECG interpretation | CODE-15% | Multi-label | 345,109 | 233,480 | 6 |
| ECG interpretation | PTB-XL(all/sub/super) | Multi-label | 21,799 | 18,869 | 71/23/5 |
| ECG interpretation[†] | PTB-XL(diag/form/rhythm) | Multi-label | 21,799 | 18,869 | 44/19/12 |
| **Pediatric ECG interpretation** | | | | | |
| ECG interpretation | ZZU pECG | Multi-label | 12,328 | 10,350 | 58 |
| **Cardiac structure & function** | | | | | |
| Echocardiogram findings | EchoNext | Multi-label | 82,543 | 36,286 | 11 |
| **Cardiac outcomes** | | | | | |
| Cardiac discharge diagnoses | MIMIC-IV-ECG | Multi-label | 114,355 | 49,400 | 158 |
| **Non-cardiac outcomes** | | | | | |
| Non-cardiac discharge diagnoses | MIMIC-IV-ECG | Multi-label | 178,163 | 81,930 | 918 |
| **Acute care predictions** | | | | | |
| Clinical deterioration | MIMIC-IV-ECG | Multi-label | 5,577 | 4,595 | 6 |
| Mortality | MIMIC-IV-ECG | Multi-label | 17,639 | 10,220 | 7 |
| ICU admission | MIMIC-IV-ECG | Multi-label | 18,690 | 13,868 | 2 |
| **Patient characteristics** | | | | | |
| Sex | MIMIC-IV-ECG | Binary | 182,076 | 83,736 | 1 |
| Age | MIMIC-IV-ECG | Regression | 182,076 | 83,736 | 1 |
| Biometrics | MIMIC-IV-ECG | Regression | 119,214 | 53,702 | 3 |
| ECG features | MIMIC-IV-ECG | Regression | 181,989 | 83,721 | 7 |
| Lab values | MIMIC-IV-ECG | Regression | 88,448 | 53,293 | 18 |
| Vital signs | MIMIC-IV-ECG | Regression | 131,602 | 66,605 | 6 |

## A.2  MODEL ARCHITECTURES

### A.2.1  ECG-CPC MODEL

The model architecture largely follows (Mehari & Strodthoff, 2023) and is composed of four convolutional layers as the encoder, followed by four S4 layers (Gu et al., 2022) (state dimension 8 and

model dimension 512) as the predictor. In contrast to (Mehari & Strodthoff, 2023), the model operates at a sampling frequency of 240 Hz, which is the minimal sampling frequency in the HEEDB dataset (Koscova et al., 2024) used for pretraining. To account for the deviation from the sampling frequency in the original publication, the model uses a kernel size of 3 and a stride of 2 in the first convolutional layer and predicts ahead 14 steps (as compared to 12 originally) in the CPC objective.

### A.2.2 S4 MODEL

The S4-based supervised baseline follows the specification in (Strodthoff et al., 2024). It also uses four S4 layers (Gu et al., 2022) (state dimension 8 and model dimension 512) without a convolutional encoder. It operates at a sampling frequency of 100 Hz with an input size of 2.5 seconds.

### A.3 PREDICTIVE PERFORMANCE

### A.3.1 FINETUNING

Quantitative finetuning results are compiled in Table 3.

Table 3: Comparison of aggregated macro-AUROC for classification and MAE for regression under finetuning with a linear prediction head. We highlight with ↑ tasks where higher AUROC is better and ↓ tasks where lower standardized MAE values are better. The best-performing result is highlighted in boldface and underlined, while models that do not perform statistically significantly worse are also highlighted in boldface. † signifies evaluation of a model trained on another dataset (listed above).

| | FMs (Finetuned) | | | | | | | | Supervised | |
| | ECGFounder | ECG-JEPA | ST-MEM | MERL | ECGFM-KED | HuBERT-ECG | ECG-FM | ECG-CPC | S4 | Net1D |
|---|---|---|---|---|---|---|---|---|---|---|
| | **Adult ECG interpretation** | | | | | | | | | |
| PTB ↑ | **0.656** | **0.679** | **0.694** | **0.717** | 0.612 | **0.699** | **0.725** | 0.702 | 0.654 | 0.564 |
| Ningbo ↑ | **0.974** | **0.973** | 0.954 | 0.955 | 0.940 | 0.958 | **0.971** | **0.973** | **0.972** | 0.968 |
| CPSC2018 ↑ | 0.966 | **0.974** | 0.946 | 0.936 | 0.930 | 0.956 | **0.972** | 0.969 | 0.962 | 0.949 |
| CPSC-Extra ↑ | **0.906** | **0.897** | **0.883** | 0.873 | 0.824 | 0.876 | 0.862 | **0.898** | 0.852 | 0.818 |
| Georgia ↑ | **0.920** | 0.918 | 0.888 | **0.912** | 0.877 | 0.883 | **0.912** | **0.913** | 0.903 | 0.884 |
| Chapman ↑ | **0.968** | **0.972** | 0.948 | 0.946 | 0.917 | 0.941 | 0.956 | 0.962 | 0.963 | 0.953 |
| -Chapman (rhythm)† ↑ | **0.991** | **0.989** | 0.985 | 0.975 | 0.963 | 0.982 | **0.993** | 0.987 | 0.986 | 0.983 |
| SPH ↑ | **0.983** | **0.980** | 0.964 | 0.944 | 0.932 | 0.953 | 0.966 | **0.981** | **0.981** | 0.967 |
| CODE-15% ↑ | 0.987 | **0.991** | 0.974 | 0.982 | 0.964 | **0.991** | 0.986 | 0.989 | 0.990 | 0.983 |
| PTB-XL (all) ↑ | 0.934 | 0.940 | 0.908 | 0.925 | 0.889 | 0.915 | 0.927 | **0.949** | 0.941 | 0.929 |
| -PTB-XL (diag) † ↑ | **0.950** | **0.946** | 0.904 | 0.942 | 0.913 | 0.925 | 0.926 | **0.951** | 0.943 | 0.940 |
| -PTB-XL (form)† ↑ | 0.875 | 0.912 | 0.887 | 0.891 | 0.829 | 0.860 | 0.904 | **0.934** | 0.919 | 0.893 |
| -PTB-XL (rhythm)† ↑ | **0.965** | **0.956** | 0.951 | 0.912 | 0.888 | 0.950 | **0.959** | **0.959** | 0.956 | 0.937 |
| PTB-XL (sub) ↑ | **0.943** | 0.935 | 0.916 | **0.937** | 0.908 | 0.918 | **0.932** | 0.940 | 0.938 | 0.926 |
| PTB-XL (super) ↑ | **0.935** | 0.921 | 0.901 | 0.930 | 0.905 | 0.908 | 0.916 | **0.934** | **0.932** | 0.924 |
| | **Pediatric ECG interpretation** | | | | | | | | | |
| ZZU pECG ↑ | 0.898 | **0.911** | 0.893 | 0.886 | 0.861 | 0.883 | 0.887 | 0.892 | 0.897 | 0.868 |
| | **Cardiac structure & function** | | | | | | | | | |
| EchoNext (Echo) ↑ | 0.817 | 0.817 | 0.816 | 0.822 | 0.806 | 0.792 | 0.772 | **0.831** | 0.819 | 0.803 |
| | **Cardiac outcomes** | | | | | | | | | |
| MIMIC (Cardiac) ↑ | 0.768 | 0.772 | 0.760 | 0.776 | 0.767 | 0.719 | 0.775 | **0.781** | **0.780** | 0.747 |
| | **Non-cardiac outcomes** | | | | | | | | | |
| MIMIC (Non-cardiac) ↑ | 0.701 | 0.711 | 0.688 | 0.712 | 0.702 | 0.642 | **0.719** | **0.719** | 0.714 | 0.672 |
| | **Acute care predictions** | | | | | | | | | |
| MIMIC (Deterioration) ↑ | 0.717 | **0.747** | 0.714 | **0.743** | 0.728 | 0.664 | **0.767** | 0.764 | **0.756** | 0.731 |
| MIMIC (Mortality) ↑ | **0.810** | 0.792 | 0.784 | **0.800** | 0.768 | 0.722 | **0.811** | 0.803 | 0.793 | 0.744 |
| MIMIC (ICU) ↑ | **0.748** | 0.742 | 0.737 | 0.744 | 0.734 | 0.710 | **0.750** | **0.753** | 0.745 | 0.721 |
| | **Patient characteristics** | | | | | | | | | |
| MIMIC (Sex) ↑ | 0.913 | 0.904 | 0.883 | 0.916 | 0.903 | 0.810 | 0.919 | **0.933** | 0.919 | 0.869 |
| MIMIC (Age) ↓ | 0.461 | 0.463 | 0.504 | 0.449 | 0.466 | 0.579 | **0.412** | 0.437 | 0.455 | 0.518 |
| MIMIC (Biometrics) ↓ | 0.637 | 0.640 | 0.673 | 0.625 | 0.647 | 0.723 | 0.620 | **0.604** | 0.626 | 0.684 |
| MIMIC (ECG Features) ↓ | 0.458 | 0.460 | 0.500 | 0.463 | 0.466 | 0.529 | 0.465 | **0.451** | 0.452 | 0.486 |
| MIMIC (Lab Values) ↓ | 0.679 | **0.677** | 0.688 | **0.676** | 0.685 | 0.709 | 0.688 | **0.673** | **0.675** | 0.691 |
| MIMIC (Vital Signs) ↓ | 0.704 | 0.703 | 0.715 | 0.702 | 0.705 | 0.717 | 0.704 | **0.700** | **0.701** | 0.712 |

### A.3.2 FROZEN EVALUATION

Frozen evaluation results are compiled in Table 4.

Table 4: Comparison of aggregated macro-AUROC for classification and MAE for regression under the frozen evaluation mode. We highlight with ↑ tasks where higher AUROC is better and ↓ tasks where lower standardized MAE values are better. The best-performing result is highlighted in boldface and underlined, while models that do not perform statistically significantly worse are also highlighted in boldface. † signifies evaluation of a model trained on another dataset (listed above).

| | FMs (Frozen Evaluation) | | | | | | | | Supervised | |
| | ECGFounder | ECG-JEPA | ST-MEM | MERL | ECGFM-KED | HuBERT-ECG | ECG-FM | ECG-CPC | S4 | Net1D |
|---|---|---|---|---|---|---|---|---|---|---|
| **Adult ECG interpretation** | | | | | | | | | | |
| PTB ↑ | **0.681** | **0.681** | **0.686** | **0.661** | 0.592 | 0.592 | 0.639 | **0.716** | 0.654 | 0.564 |
| Ningbo ↑ | 0.961 | **0.971** | 0.956 | 0.942 | 0.833 | 0.911 | 0.927 | 0.953 | **0.972** | **0.968** |
| CPSC2018 ↑ | 0.966 | **0.975** | 0.956 | 0.956 | 0.874 | 0.919 | 0.930 | 0.959 | 0.962 | 0.949 |
| CPSC-Extra ↑ | **0.907** | **0.902** | 0.867 | 0.872 | 0.739 | 0.831 | 0.840 | 0.887 | 0.852 | 0.818 |
| Georgia ↑ | **0.924** | **0.910** | 0.893 | 0.890 | 0.787 | 0.836 | 0.858 | 0.894 | 0.903 | 0.884 |
| Chapman ↑ | **0.967** | **0.964** | 0.955 | 0.954 | 0.852 | 0.917 | 0.900 | 0.943 | **0.963** | 0.953 |
| -Chapman (rhythm)† ↑ | **0.983** | **0.988** | **0.984** | 0.972 | 0.849 | 0.951 | 0.974 | 0.983 | **0.986** | **0.983** |
| SPH ↑ | 0.966 | **0.980** | 0.946 | 0.958 | 0.885 | 0.939 | 0.938 | 0.961 | **0.981** | 0.967 |
| CODE-15% ↑ | 0.980 | **0.990** | 0.965 | 0.879 | 0.688 | 0.978 | 0.968 | 0.983 | **0.990** | **0.983** |
| PTB-XL (all) ↑ | 0.927 | 0.934 | 0.910 | 0.909 | 0.810 | 0.883 | 0.884 | 0.931 | **0.941** | 0.929 |
| -PTB-XL (diag) † ↑ | **0.940** | **0.943** | 0.910 | 0.918 | 0.827 | 0.898 | 0.890 | 0.934 | **0.943** | **0.940** |
| -PTB-XL (form) † ↑ | 0.876 | 0.889 | 0.894 | 0.879 | 0.783 | 0.815 | 0.838 | **0.905** | **0.919** | 0.893 |
| -PTB-XL (rhythm) † ↑ | **0.958** | **0.966** | 0.934 | 0.928 | 0.792 | 0.917 | 0.931 | 0.951 | **0.956** | 0.937 |
| PTB-XL (sub) ↑ | **0.939** | **0.934** | **0.931** | 0.917 | 0.816 | 0.903 | 0.912 | **0.934** | **0.938** | 0.926 |
| PTB-XL (super) ↑ | 0.928 | 0.917 | 0.923 | 0.915 | 0.865 | 0.892 | 0.877 | 0.919 | **0.932** | 0.924 |
| **Pediatric ECG interpretation** | | | | | | | | | | |
| ZZU pECG ↑ | 0.891 | **0.905** | **0.899** | 0.870 | 0.789 | 0.857 | 0.845 | 0.879 | **0.897** | 0.868 |
| **Cardiac structure & function** | | | | | | | | | | |
| EchoNext (Echo) ↑ | 0.803 | 0.811 | **0.817** | 0.801 | 0.791 | 0.778 | 0.772 | **0.822** | **0.819** | 0.803 |
| **Cardiac outcomes** | | | | | | | | | | |
| MIMIC (Cardiac) ↑ | 0.745 | 0.757 | 0.734 | 0.755 | 0.708 | 0.736 | 0.688 | 0.774 | **0.780** | 0.747 |
| **Non-cardiac outcomes** | | | | | | | | | | |
| MIMIC (Non-cardiac) ↑ | 0.671 | 0.688 | 0.657 | 0.681 | 0.630 | 0.659 | 0.616 | 0.703 | **0.714** | 0.672 |
| **Acute care predictions** | | | | | | | | | | |
| MIMIC (Deterioration) ↑ | 0.697 | 0.702 | 0.648 | 0.721 | 0.661 | 0.685 | 0.639 | 0.743 | **0.756** | **0.731** |
| MIMIC (Mortality) ↑ | **0.769** | **0.788** | **0.754** | 0.761 | 0.720 | 0.723 | 0.731 | **0.785** | **0.793** | 0.744 |
| MIMIC (ICU) ↑ | 0.731 | 0.734 | 0.715 | 0.727 | 0.698 | 0.719 | 0.691 | **0.750** | **0.745** | 0.721 |
| **Patient characteristics** | | | | | | | | | | |
| MIMIC (Sex) ↑ | 0.872 | 0.894 | 0.883 | 0.879 | 0.839 | 0.841 | 0.826 | **0.918** | **0.919** | 0.869 |
| MIMIC (Age) ↓ | 0.515 | 0.484 | 0.501 | 0.511 | 0.581 | 0.544 | 0.542 | 0.466 | **0.455** | 0.518 |
| MIMIC (Biometrics) ↓ | 0.702 | 0.700 | 0.681 | 0.683 | 0.715 | 0.702 | 0.751 | 0.640 | **0.626** | 0.684 |
| MIMIC (ECG Features) ↓ | 0.489 | 0.477 | 0.489 | 0.507 | 0.563 | 0.500 | 0.566 | 0.465 | **0.452** | 0.486 |
| MIMIC (Lab Values) ↓ | 0.703 | 0.694 | 0.740 | 0.694 | 0.712 | 0.703 | 0.763 | **0.676** | **0.675** | 0.691 |
| MIMIC (Vital Signs) ↓ | 0.719 | 0.716 | 0.739 | 0.716 | 0.729 | 0.722 | 0.747 | 0.703 | **0.701** | 0.712 |

### A.3.3 LINEAR EVALUATION

Linear evaluation results are compiled in Table 5.

### A.3.4 RANKING

In Table 6, we summarize model performance for each task in terms of a ranked list with ties, accounting for statistical significance. In Table 7, we summarize these results by reporting the median ranks for each of the seven categories considered.

### A.4 LABEL-SPECIFIC MODEL PREDICTIONS

In Table 8-Table 29, we show model predictions for the 10 best predicted labels (sorted by the performance of the supervised S4 model). These tables reflect the high degree of specificity of the tasks that are covered by this benchmark.

### A.5 SCALING ANALYSIS

Table 30 lists the fit parameters for the scaling curves of the form $CN^{-\alpha} + L_0$. Taking the supervised baseline (S4) as reference, one can use these parametric forms to work out for a given training set size $N$, what the required training set size would be $N^*$ to reach the same level of performance with a given pretrained model. The ratio $r = N^*/N$ then characterizes the improved label efficiency of the pretrained model. For each of the models, we evaluate $r$ for different training dataset sizes. The results are compiled in Table 31.

Table 5: Comparison of aggregated macro-AUROC for classification and MAE for regression under the linear evaluation mode. We highlight with ↑ tasks where higher AUROC is better and ↓ tasks where lower standardized MAE values are better. The best-performing result is highlighted in boldface and underlined, while models that do not perform statistically significantly worse are also highlighted in boldface. † signifies evaluation of a model trained on another dataset (listed above).

| | FMs (Linear Evaluation) | | | | | | | | Supervised | |
|---|---|---|---|---|---|---|---|---|---|---|
| | ECGFounder | ECG-JEPA | ST-MEM | MERL | ECGFM-KED | HuBERT-ECG | ECG-FM | ECG-CPC | S4 | Net1D |
| **Adult ECG interpretation** | | | | | | | | | | |
| PTB ↑ | **0.671** | **0.665** | **0.692** | 0.583 | 0.503 | **0.604** | **0.692** | 0.578 | **0.654** | 0.564 |
| Ningbo ↑ | **0.970** | **0.970** | 0.954 | 0.916 | 0.762 | 0.896 | 0.902 | 0.898 | **0.972** | **0.968** |
| CPSC2018 ↑ | 0.964 | **0.975** | 0.945 | 0.914 | 0.786 | 0.899 | 0.906 | 0.902 | 0.962 | 0.949 |
| CPSC-Extra ↑ | **0.910** | **0.902** | 0.885 | 0.858 | 0.553 | 0.855 | 0.842 | 0.794 | 0.852 | 0.818 |
| Georgia ↑ | **0.923** | 0.920 | 0.889 | 0.872 | 0.642 | 0.847 | 0.847 | 0.854 | 0.903 | 0.884 |
| Chapman ↑ | **0.968** | **0.962** | 0.949 | 0.916 | 0.745 | 0.904 | 0.891 | 0.868 | **0.963** | 0.953 |
| -Chapman (rhythm)† ↑ | **0.987** | **0.989** | 0.985 | 0.946 | 0.776 | 0.953 | 0.968 | 0.944 | **0.986** | 0.983 |
| SPH ↑ | 0.975 | 0.967 | 0.966 | 0.943 | 0.798 | 0.894 | 0.914 | 0.928 | **0.981** | 0.967 |
| CODE-15% ↑ | 0.976 | **0.984** | **0.975** | 0.716 | 0.568 | 0.965 | 0.976 | 0.968 | **0.990** | 0.983 |
| PTB-XL (all) ↑ | 0.931 | 0.928 | 0.908 | 0.883 | 0.706 | 0.867 | 0.841 | 0.904 | **0.941** | 0.929 |
| -PTB-XL (diag)† ↑ | **0.947** | 0.925 | 0.903 | 0.894 | 0.715 | 0.875 | 0.861 | 0.907 | **0.943** | 0.940 |
| -PTB-XL (form)† ↑ | 0.874 | **0.908** | 0.889 | 0.864 | 0.716 | 0.817 | 0.782 | 0.873 | **0.919** | 0.893 |
| -PTB-XL (rhythm)† ↑ | **0.961** | **0.969** | 0.951 | 0.877 | 0.666 | 0.902 | 0.865 | 0.938 | 0.956 | 0.937 |
| PTB-XL (sub) ↑ | **0.945** | 0.916 | 0.916 | 0.895 | 0.734 | 0.898 | 0.866 | 0.886 | **0.938** | 0.926 |
| PTB-XL (super) ↑ | 0.924 | 0.911 | 0.896 | 0.896 | 0.802 | 0.877 | 0.873 | 0.863 | **0.932** | 0.924 |
| **Pediatric ECG interpretation** | | | | | | | | | | |
| ZZU pECG ↑ | **0.900** | **0.891** | **0.893** | 0.847 | 0.591 | 0.827 | 0.813 | 0.852 | **0.897** | 0.868 |
| **Cardiac structure & function** | | | | | | | | | | |
| EchoNext ↑ | 0.795 | 0.806 | **0.816** | 0.794 | 0.770 | 0.770 | 0.767 | 0.800 | **0.819** | 0.803 |
| **Cardiac outcomes** | | | | | | | | | | |
| MIMIC (Cardiac) ↑ | 0.751 | 0.751 | **0.761** | 0.751 | 0.683 | 0.719 | 0.675 | 0.751 | **0.780** | 0.747 |
| **Non-cardiac outcomes** | | | | | | | | | | |
| MIMIC (Non-cardiac) ↑ | 0.671 | 0.675 | **0.688** | 0.672 | 0.617 | 0.642 | 0.599 | 0.680 | **0.714** | 0.672 |
| **Acute care predictions** | | | | | | | | | | |
| MIMIC (Deterioration) ↑ | 0.713 | 0.720 | 0.717 | 0.704 | 0.627 | 0.664 | 0.616 | **0.733** | **0.756** | **0.731** |
| MIMIC (Mortality) ↑ | **0.774** | **0.782** | **0.788** | 0.744 | 0.681 | 0.722 | 0.676 | **0.769** | **0.793** | **0.744** |
| MIMIC (ICU) ↑ | 0.730 | **0.736** | **0.737** | 0.722 | 0.658 | 0.710 | 0.683 | 0.733 | **0.745** | 0.721 |
| **Patient characteristics** | | | | | | | | | | |
| MIMIC (Sex) ↑ | 0.872 | 0.883 | 0.882 | 0.853 | 0.801 | 0.810 | 0.853 | 0.873 | **0.919** | 0.869 |
| MIMIC (Age) ↓ | 0.511 | 0.489 | 0.503 | 0.561 | 0.639 | 0.579 | 0.577 | 0.551 | **0.455** | 0.518 |
| MIMIC (Biometrics) ↓ | 0.700 | 0.685 | 0.674 | 0.711 | 0.740 | 0.723 | 0.837 | 0.686 | **0.626** | 0.684 |
| MIMIC (ECG Features) ↓ | 0.488 | 0.490 | 0.499 | 0.543 | 0.606 | 0.529 | 0.617 | 0.504 | **0.452** | 0.486 |
| MIMIC (Lab Values) ↓ | 0.693 | 0.695 | 0.690 | 0.698 | 0.713 | 0.709 | 0.930 | 0.688 | **0.675** | 0.691 |
| MIMIC (Vital Signs) ↓ | 0.716 | 0.714 | 0.711 | 0.719 | 0.738 | 0.717 | 0.834 | 0.708 | **0.701** | 0.712 |

## A.6 MODEL COMPLEXITY

In Table 32, we compare different notions of model complexity, namely, number of model parameters, number of floating point operations, peak GPU memory usage and inference throughput.

## A.7 IMPACT OF LAYER-DEPENDENT LEARNING RATES DURING FINETUNING

In Table 34, we investigate the effect of layer-dependent learning rates during finetuning across three datasets, PTB-XL, EchoNext and CPSC-Extra.

## A.8 BACKBONE COMPARISON FOR SUPERVISED TRAINING FROM SCRATCH

In Table 34, we provide additional results on the relative performance of different model backbone across three datasets, PTB-XL, EchoNext and CPSC-Extra. In addition to the S4-model, a structured state space model, and Net1D, a CNN, we consider Mamba-1 (Gu & Dao, 2024) and Mamba-2 (Dao & Gu, 2024) as an examples for more recent state-space model. More specifically, for the two Mamba models we use the following configuration d_model = 512, d_state = 128, n_layers = 4. The S4 backbone performs best or consistent with the best-performing model across all three datasets. Net1D is competitive only on a single dataset. Mamba-1 is the best perfoming model on a single dataset. This underscores the superiority of the S4-based baseline, in line with recent claims in the literature (Somvanshi et al., 2025; Patro & Agneeswaran, 2025) putting into question the suitability of latest state space models for the processing of physiological time series.

Table 6: Statistical ranking of FMs across evaluation modes and datasets. Rankings (Fine-tuned/Frozen/Linear) are assigned based on statistical equivalence groups determined by bootstrap testing, where models not performing significantly worse than the best model share the same rank. Lower ranks indicate better performance. $^\dagger$ signifies evaluation of a model trained on another dataset (listed above).

| | FMs (Finetuned/Frozen/Linear) | | | | | | | | Supervised | |
| | ECGFounder | ECG-JEPA | ST-MEM | MERL | ECGFM-KED | HuBERT-ECG | ECG-FM | ECG-CPC | S4 | Net1D |
|---|---|---|---|---|---|---|---|---|---|---|
| **Adult ECG interpretation** | | | | | | | | | | |
| PTB | 1/1/1 | 1/1/1 | 1/1/1 | 1/1/6 | 8/9/10 | 1/6/6 | 1/6/1 | 1/1/6 | 8/6/1 | 10/9/6 |
| Ningbo | 1/4/1 | 1/1/1 | 7/4/5 | 7/7/6 | 10/10/10 | 7/8/6 | 1/8/6 | 1/1/1 | 6/1/1 |
| CPSC2018 | 4/2/2 | 1/1/1 | 6/6/4 | 9/2/6 | 9/10/10 | 6/8/8 | 1/8/6 | 1/2/8 | 4/2/2 | 6/6/4 |
| CPSC-Extra | 1/1/1 | 1/1/1 | 1/3/3 | 5/3/5 | 9/10/10 | 5/6/5 | 5/6/5 | 1/3/9 | 5/6/3 | 9/6/5 |
| Georgia | 1/1/1 | 1/1/1 | 7/3/4 | 1/3/6 | 7/10/10 | 7/8/6 | 1/3/6 | 6/3/3 | 7/7/4 |
| Chapman | 1/1/1 | 1/1/1 | 6/4/4 | 6/4/6 | 10/10/10 | 9/8/6 | 3/8/8 | 3/7/8 | 3/1/1 | 6/4/4 |
| -Chapman (rhythm)$^\dagger$ | 1/1/1 | 1/1/1 | 4/1/4 | 9/8/8 | 10/10/10 | 4/9/6 | 1/6/6 | 4/6/8 | 4/1/1 | 8/1/4 |
| SPH | 1/3/2 | 1/1/2 | 5/3/5 | 5/7/6 | 10/10/10 | 5/8/7 | 5/8/7 | 1/3/7 | 1/1/1 | 5/3/2 |
| CODE-15% | 5/5/4 | 1/1/1 | 5/5/1 | 5/9/9 | 10/10/10 | 1/5/6 | 5/5/6 | 1/1/6 | 1/1/1 | 5/1/4 |
| PTB-XL (all) | 2/2/2 | 2/2/2 | 8/6/5 | 5/6/7 | 10/10/10 | 8/8/8 | 5/8/9 | 1/2/5 | 2/1/1 | 5/2/2 |
| -PTB-XL (diag)$^\dagger$ | 1/1/1 | 1/1/4 | 9/6/5 | 4/6/7 | 9/10/10 | 7/8/8 | 7/8/8 | 1/5/5 | 4/1/1 | 4/1/3 |
| -PTB-XL (form)$^\dagger$ | 7/3/3 | 2/3/1 | 7/3/3 | 4/3/7 | 9/10/10 | 9/8/8 | 4/8/8 | 1/1/3 | 2/1/1 | 4/3/3 |
| -PTB-XL (rhythm)$^\dagger$ | 1/1/1 | 1/1/1 | 5/4/3 | 9/7/7 | 10/10/10 | 5/7/7 | 1/4/9 | 1/4/5 | 5/1/3 | 8/7/5 |
| PTB-XL (sub) | 1/1/1 | 6/1/3 | 7/1/3 | 1/6/6 | 10/10/10 | 7/8/6 | 1/8/9 | 1/1/8 | 1/1/1 | 7/6/3 |
| PTB-XL (super) | 1/2/2 | 5/6/4 | 10/2/5 | 4/6/6 | 8/10/10 | 8/8/7 | 7/9/7 | 1/4/9 | 1/1/1 | 5/4/2 |
| **Pediatric ECG interpretation** | | | | | | | | | | |
| ZZU pECG | 2/4/1 | 1/1/1 | 2/1/1 | 2/6/7 | 9/10/10 | 7/6/7 | 7/9/9 | 2/4/5 | 2/1/1 | 9/6/5 |
| **Cardiac structure & function** | | | | | | | | | | |
| EchoNext (Echo) | 2/4/6 | 2/4/3 | 2/1/1 | 2/7/6 | 7/8/8 | 9/9/8 | 10/9/8 | 1/1/3 | 2/1/1 | 7/4/3 |
| **Cardiac outcomes** | | | | | | | | | | |
| MIMIC (Cardiac) | 6/5/3 | 3/3/3 | 8/7/2 | 3/3/3 | 6/9/9 | 10/7/8 | 3/10/9 | 1/2/3 | 1/1/1 | 9/5/3 |
| **Non-cardiac outcomes** | | | | | | | | | | |
| MIMIC (Non-cardiac) | 6/5/7 | 4/3/4 | 8/7/2 | 4/4/4 | 6/9/9 | 10/7/8 | 1/10/10 | 1/2/3 | 3/1/1 | 9/5/4 |
| **Acute care predictions** | | | | | | | | | | |
| MIMIC (Deterioration) | 6/4/5 | 1/4/1 | 6/8/5 | 1/4/5 | 6/8/8 | 10/4/8 | 1/8/8 | 1/1/1 | 1/1/1 | 6/1/1 |
| MIMIC (Mortality) | 1/1/1 | 1/1/1 | 1/1/1 | 1/6/7 | 8/6/7 | 8/10/7 | 1/6/7 | 1/1/1 | 1/1/1 | 8/6/1 |
| MIMIC (ICU) | 1/3/4 | 4/3/1 | 4/6/1 | 4/3/6 | 8/9/10 | 10/6/6 | 1/9/9 | 1/1/4 | 4/1/1 | 9/6/6 |
| **Patient characteristics** | | | | | | | | | | |
| MIMIC (Sex) | 5/6/4 | 6/3/2 | 8/4/2 | 2/4/7 | 6/8/9 | 10/8/9 | 2/10/7 | 1/1/4 | 2/1/1 | 9/6/4 |
| MIMIC (Age) | 5/5/4 | 5/3/2 | 8/4/3 | 3/5/7 | 5/10/10 | 10/8/8 | 1/8/8 | 2/2/6 | 4/1/1 | 9/7/5 |
| MIMIC (Biometrics) | 5/6/6 | 5/6/3 | 8/3/2 | 2/3/7 | 7/9/9 | 10/6/8 | 2/10/10 | 1/2/3 | 2/1/1 | 9/3/3 |
| MIMIC (ECG Features) | 3/5/3 | 4/3/4 | 9/5/5 | 5/8/8 | 7/9/9 | 10/7/7 | 5/9/10 | 1/2/6 | 2/1/1 | 8/4/2 |
| MIMIC (Lab Values) | 5/6/4 | 1/3/6 | 6/9/2 | 1/5/6 | 6/8/9 | 10/6/8 | 6/10/10 | 1/1/2 | 1/1/1 | 9/3/4 |
| MIMIC (Vital Signs) | 6/4/5 | 3/4/5 | 9/9/3 | 3/4/8 | 6/8/9 | 9/7/7 | 3/10/10 | 1/2/2 | 1/1/1 | 8/3/3 |

Table 7: Median statistical rankings of FMs across evaluation modes by categories. Rankings (Fine-tuned/Frozen/Linear) represent the median performance position across all datasets within each category. Lower values indicate better overall performance.

| | FMs (Finetuned/Frozen/Linear) | | | | | | | | Supervised | |
| | ECGFounder | ECG-JEPA | ST-MEM | MERL | ECGFM-KED | HuBERT-ECG | ECG-FM | ECG-CPC | S4 | Net1D |
|---|---|---|---|---|---|---|---|---|---|---|
| Adult ECG interpretation | 1/1/1 | 1/1/1 | 6/3/4 | 5/6/6 | 10/10/10 | 7/8/7 | 3/8/7 | 1/3/7 | 3/1/1 | 6/4/4 |
| Pediatric ECG interpretation | 2/4/1 | 1/1/1 | 2/1/1 | 2/6/7 | 9/10/10 | 7/6/7 | 7/9/9 | 2/4/5 | 2/1/1 | 9/6/5 |
| Cardiac structure & function | 2/4/6 | 2/4/3 | 2/1/1 | 2/7/6 | 7/8/8 | 9/9/8 | 10/9/8 | 1/1/3 | 2/1/1 | 7/4/3 |
| Cardiac outcomes | 6/5/3 | 3/3/3 | 8/7/2 | 3/3/3 | 6/9/9 | 10/7/8 | 3/10/9 | 1/2/3 | 1/1/1 | 9/5/3 |
| Non-cardiac outcomes | 6/5/7 | 4/3/4 | 8/7/2 | 4/4/4 | 6/9/9 | 10/7/8 | 1/10/10 | 1/2/3 | 3/1/1 | 9/5/4 |
| Acute care predictions | 1/3/4 | 1/3/1 | 4/6/1 | 1/4/6 | 8/8/8 | 10/6/7 | 1/8/8 | 1/1/1 | 1/1/1 | 8/6/1 |
| Patient characteristics | 5/5.5/4 | 4.5/3/3.5 | 8/4.5/2.5 | 2.5/4.5/7 | 6/8.5/9 | 10/7/8 | 2.5/9.5/10 | 1/2/3.5 | 2/1/1 | 9/3.5/3.5 |

## A.9 Representational Analysis Based on CKA

### A.9.1 Intra-model Analysis

We start by describing the intra-model CKA patterns in detail. We investigate the FMs in their original form without any task-specific finetuning. The results are compiled in Figure 4.

**ECG-CPC** In Figure 4a, the early CNN layers (CNN0-CNN2) show a very high similarity suggesting that they learn very similar representations, which could be taken as a sign that the CNN encoder is potentially overparameterized for the task. The final CNN layer (CNN3) shows similarity with both the earlier CNN layer as well as the first SSM layer (S4-0) thereby serving as a transition point between the local, CNN-based feature extractor and the sequential SSM layers in the hybrid

Table 8: Finetuning with a linear prediction head performance for the 10 best-predicted labels, sorted by supervised S4 AUROC on the PTB dataset.

| Label | ECGFounder | ECG-JEPA | ECGFM-KED | MERL | ST-MEM | HuBERT-ECG | ECG-FM | ECG-CPC | S4 | Net1D |
|---|---|---|---|---|---|---|---|---|---|---|
| Healthy control | **0.968** | 0.919 | 0.901 | 0.917 | 0.833 | 0.922 | 0.911 | 0.884 | 0.881 | 0.920 |
| Arterial Hypertension | 0.638 | 0.400 | 0.724 | 0.705 | 0.810 | 0.610 | **0.924** | 0.619 | 0.876 | 0.352 |
| Atrial fibrillation | 0.997 | 0.958 | **1.000** | 0.968 | 0.430 | 0.955 | **1.000** | 0.893 | 0.780 | 0.372 |
| Cardiomyopathy | 0.812 | 0.880 | 0.913 | 0.731 | **0.923** | 0.909 | 0.740 | 0.875 | 0.764 | 0.822 |
| Ventricular fibrillation | 0.465 | 0.684 | 0.528 | 0.786 | 0.547 | 0.709 | 0.584 | 0.564 | **0.736** | 0.664 |
| Myocardial infarction acute | 0.686 | 0.330 | 0.706 | **0.819** | 0.809 | 0.414 | 0.650 | 0.793 | 0.722 | 0.731 |
| Myocardial infarction old | 0.790 | **0.810** | 0.767 | 0.806 | 0.675 | 0.758 | 0.746 | 0.727 | 0.717 | 0.696 |
| Myocardial infarction acute catheterized | 0.778 | 0.760 | 0.828 | 0.736 | 0.652 | 0.767 | **0.824** | 0.710 | 0.715 | 0.696 |
| Obesity | 0.498 | 0.755 | 0.682 | 0.557 | 0.668 | 0.430 | 0.647 | **0.795** | 0.700 | 0.507 |
| Arterial hypertension | 0.552 | 0.713 | 0.621 | 0.743 | 0.654 | 0.704 | 0.674 | 0.680 | **0.693** | 0.556 |

Table 9: Finetuning with a linear prediction head performance for the 10 best-predicted labels, sorted by supervised S4 AUROC on the Ningbo dataset.

| Label | ECGFounder | ECG-JEPA | ECGFM-KED | MERL | ST-MEM | HuBERT-ECG | ECG-FM | ECG-CPC | S4 | Net1D |
|---|---|---|---|---|---|---|---|---|---|---|
| Wolff-Parkinson-White | **1.000** | **1.000** | **1.000** | **1.000** | 0.997 | 0.998 | **1.000** | **1.000** | **1.000** | **1.000** |
| AAR | 0.997 | 0.997 | 0.999 | **1.000** | 0.996 | 0.898 | 0.999 | 0.997 | 0.999 | 0.998 |
| Ventricular fibrillation | **1.000** | **1.000** | **1.000** | 0.994 | 0.990 | **1.000** | **1.000** | 0.999 | 0.999 | 0.999 |
| Atrial flutter | **0.999** | **0.999** | 0.998 | 0.998 | 0.996 | 0.997 | **0.999** | 0.998 | **0.999** | 0.998 |
| Right atrial hypertrophy | 0.996 | 0.999 | 0.984 | **1.000** | **1.000** | 0.905 | 0.994 | 0.997 | 0.998 | 0.998 |
| Ventricular escape rhythm | **0.999** | **0.999** | **0.999** | 0.998 | 0.998 | **0.999** | 0.998 | **0.999** | 0.998 | **0.999** |
| Sinus tachycardia | **0.998** | 0.997 | 0.995 | 0.996 | 0.995 | 0.996 | 0.997 | **0.998** | **0.998** | 0.997 |
| Junctional tachycardia | 0.985 | **1.000** | 0.996 | 0.972 | 0.997 | **1.000** | 0.999 | 0.992 | 0.998 | 0.997 |
| Sinus bradycardia | 0.998 | 0.998 | 0.996 | 0.994 | 0.998 | 0.998 | **0.999** | 0.998 | 0.998 | 0.998 |
| Complete left bundle branch block | **0.999** | 0.998 | 0.998 | **0.999** | 0.998 | **0.999** | **0.999** | 0.998 | 0.997 | 0.998 |

Table 10: Finetuning with a linear prediction head performance for the 10 best-predicted labels, sorted by supervised S4 AUROC on the CPSC2018 dataset.

| Label | ECGFounder | ECG-JEPA | ECGFM-KED | MERL | ST-MEM | HuBERT-ECG | ECG-FM | ECG-CPC | S4 | Net1D |
|---|---|---|---|---|---|---|---|---|---|---|
| Left bundle branch block | 0.998 | **0.999** | 0.998 | **0.999** | 0.998 | **0.999** | **0.999** | 0.998 | 0.998 | 0.998 |
| 1st degree atrioventricular block | **0.990** | **0.990** | 0.971 | 0.963 | 0.946 | 0.981 | 0.980 | 0.982 | 0.987 | 0.982 |
| Atrial fibrillation | 0.994 | 0.980 | 0.993 | 0.982 | 0.978 | 0.991 | **0.995** | 0.988 | 0.985 | 0.982 |
| Right bundle branch block | **0.989** | 0.981 | 0.971 | 0.983 | 0.974 | 0.981 | 0.983 | 0.983 | 0.985 | 0.980 |
| NORMAL | 0.966 | **0.973** | 0.954 | 0.948 | 0.930 | 0.947 | 0.963 | **0.973** | 0.961 | 0.961 |
| ST depression | 0.963 | **0.974** | 0.945 | 0.947 | 0.948 | 0.932 | 0.963 | 0.973 | 0.961 | 0.957 |
| Premature ventricular contraction | 0.941 | 0.957 | 0.945 | 0.891 | 0.874 | 0.914 | **0.969** | 0.946 | 0.948 | 0.931 |
| ST elevation | 0.925 | **0.962** | 0.849 | 0.885 | 0.887 | 0.922 | 0.938 | 0.951 | 0.943 | 0.873 |
| Premature atrial contraction | 0.927 | 0.950 | 0.884 | 0.828 | 0.835 | 0.941 | **0.957** | 0.928 | 0.890 | 0.881 |

Table 11: Finetuning with a linear prediction head performance for the 10 best-predicted labels, sorted by supervised S4 AUROC on the CPSC-Extra dataset.

| Label | ECGFounder | ECG-JEPA | ECGFM-KED | MERL | ST-MEM | HuBERT-ECG | ECG-FM | ECG-CPC | S4 | Net1D |
|---|---|---|---|---|---|---|---|---|---|---|
| Right atrial hypertrophy | **1.000** | 0.979 | 0.993 | 0.956 | 0.979 | 0.996 | 0.975 | 0.929 | **1.000** | 0.912 |
| Complete heart block | 0.999 | **1.000** | 0.999 | 0.999 | 0.987 | 0.997 | **1.000** | **1.000** | 0.996 | 0.999 |
| 2nd degree atrioventricular block | 0.997 | **1.000** | 0.996 | 0.996 | 0.972 | **1.000** | **1.000** | 0.996 | 0.994 | 0.994 |
| Complete right bundle branch block | 0.994 | 0.994 | 0.985 | 0.994 | 0.989 | **0.995** | 0.988 | 0.994 | 0.988 | 0.973 |
| Sinus tachycardia | 0.986 | 0.988 | **0.990** | 0.978 | 0.922 | 0.989 | 0.987 | **0.990** | 0.985 | 0.987 |
| Atrial fibrillation and flutter | 0.990 | 0.986 | 0.984 | 0.990 | 0.913 | 0.969 | 0.983 | **0.996** | 0.979 | 0.977 |
| Bradycardia | 0.978 | 0.975 | 0.970 | 0.970 | 0.932 | 0.976 | 0.942 | **0.981** | 0.978 | 0.980 |
| Atrial flutter | 0.985 | **0.990** | **0.990** | **0.990** | 0.919 | 0.948 | 0.902 | 0.975 | 0.977 | 0.962 |
| Incomplete right bundle branch block | 0.982 | 0.976 | 0.978 | 0.977 | 0.941 | 0.971 | 0.965 | **0.986** | 0.975 | 0.940 |
| Atrial tachycardia | 0.990 | 0.959 | 0.991 | 0.976 | 0.926 | **0.997** | 0.991 | 0.978 | 0.972 | 0.516 |

architecture. The still pronounced dissimilarity between the the final CNN layer (CNN3) and the first SSM layer (S4-0) can most likely be explained by the difference in model architecture between these two layers. The following S4 layers become increasingly dissimilar to the CNN encoder layers, suggesting a substantial evolution throughout the architecture. Finally, the last S4 layer (S4-3) is highly specialized and shows only moderate to low similarity to other S4 layers, suggesting it learns the most task-specific representations for the CPC task. Overall the model shows a clear representational gradient from low-level CNN-features to high-level sequential features.

**ECGFounder** In Figure 4b, we show the CKA analysis for the ECGFounder model. The convolutional stem shows low similarity with all other layers. Apart from that, the CNN model shows a gradual transition across layers with no abrupt transitions as observed in the hybrid ECG-CPC model between the convolutional feature extractor and the S4 layers. The model shows highly similar rep-

Table 12: Finetuning with a linear prediction head performance for the 10 best-predicted labels, sorted by supervised S4 AUROC on the Georgia dataset.

| Label | ECGFounder | ECG-JEPA | ECGFM-KED | MERL | ST-MEM | HuBERT-ECG | ECG-FM | ECG-CPC | S4 | Net1D |
|---|---|---|---|---|---|---|---|---|---|---|
| 2nd degree atrioventricular block | 0.999 | 0.997 | 0.999 | 0.998 | 0.998 | 0.997 | 0.998 | **1.000** | **1.000** | **1.000** |
| Left bundle branch block | **0.998** | **0.998** | **0.998** | 0.997 | 0.991 | **0.998** | 0.997 | **0.998** | 0.997 | **0.998** |
| Sinus bradycardia | 0.997 | 0.994 | 0.996 | 0.989 | 0.979 | 0.996 | 0.990 | 0.996 | 0.996 | **0.998** |
| Sinus tachycardia | **0.995** | 0.988 | 0.988 | 0.981 | 0.973 | 0.994 | 0.984 | 0.994 | 0.993 | 0.991 |
| Supraventricular tachycardia | 0.995 | 0.990 | 0.994 | **1.000** | 0.985 | 0.996 | 0.978 | 0.995 | 0.988 | 0.958 |
| Ventricular pacing pattern | 0.922 | 0.981 | **0.987** | 0.972 | 0.982 | 0.968 | 0.908 | 0.979 | 0.982 | 0.725 |
| Left anterior fascicular block | **0.986** | 0.974 | 0.880 | 0.978 | 0.956 | 0.965 | 0.963 | 0.971 | 0.981 | 0.982 |
| Right bundle branch block | **0.992** | 0.980 | 0.987 | 0.982 | 0.982 | 0.986 | 0.986 | 0.986 | 0.980 | 0.982 |
| Bundle branch block | 0.978 | **0.981** | 0.941 | 0.960 | 0.951 | 0.962 | 0.928 | 0.969 | 0.976 | 0.971 |
| 1st degree atrioventricular block | **0.984** | 0.980 | **0.984** | 0.975 | 0.944 | 0.943 | 0.983 | 0.983 | 0.976 | 0.970 |

Table 13: Finetuning with a linear prediction head performance for the 10 best-predicted labels, sorted by supervised S4 AUROC on the Chapman dataset.

| Label | ECGFounder | ECG-JEPA | ECGFM-KED | MERL | ST-MEM | HuBERT-ECG | ECG-FM | ECG-CPC | S4 | Net1D |
|---|---|---|---|---|---|---|---|---|---|---|
| Myocardial infarction in the lower wall | 0.999 | 0.997 | **1.000** | 0.997 | 0.995 | **1.000** | 0.999 | **1.000** | **1.000** | 0.998 |
| Sinus bradycardia | **1.000** | **1.000** | 0.999 | 0.996 | 0.994 | 0.999 | 0.999 | 0.999 | 0.999 | 0.999 |
| Sinus tachycardia | **0.999** | 0.998 | 0.994 | 0.995 | 0.991 | 0.996 | **0.999** | **0.999** | 0.998 | 0.997 |
| Myocardial infarction in the front wall | 0.997 | 0.995 | 0.994 | 0.998 | **0.999** | 0.993 | 0.997 | 0.997 | 0.997 | 0.987 |
| Supraventricular tachycardia | **0.998** | 0.997 | 0.996 | 0.994 | 0.983 | 0.994 | 0.998 | 0.997 | 0.996 | 0.997 |
| Long RR interval | 0.997 | **0.999** | **0.999** | 0.995 | 0.954 | 0.996 | 0.994 | 0.996 | 0.996 | 0.995 |
| Atrial fibrillation | 0.997 | 0.996 | 0.996 | 0.994 | 0.987 | 0.992 | **0.998** | 0.996 | 0.994 | 0.992 |
| Atrial flutter | 0.990 | 0.984 | 0.987 | 0.944 | 0.909 | 0.978 | **0.996** | 0.992 | 0.993 | 0.992 |
| Left front bundle branch block | 0.989 | 0.981 | 0.934 | 0.989 | 0.989 | 0.985 | 0.974 | 0.979 | 0.992 | **0.994** |
| Right bundle-branch block | **0.996** | 0.995 | 0.990 | 0.993 | 0.989 | 0.993 | 0.994 | 0.995 | 0.991 | 0.994 |

Table 14: Finetuning with a linear prediction head performance for the 10 best-predicted labels, sorted by supervised S4 AUROC on the SPH dataset.

| Label | ECGFounder | ECG-JEPA | ECGFM-KED | MERL | ST-MEM | HuBERT-ECG | ECG-FM | ECG-CPC | S4 | Net1D |
|---|---|---|---|---|---|---|---|---|---|---|
| AV block, complete (third-degree) | **1.000** | **1.000** | **1.000** | **1.000** | 0.994 | **1.000** | **1.000** | **1.000** | **1.000** | **1.000** |
| Left bundle-branch block | **1.000** | **1.000** | 0.999 | **1.000** | **1.000** | **1.000** | **1.000** | **1.000** | **1.000** | 0.999 |
| Junctional escape complex(es) | **1.000** | **1.000** | **1.000** | 0.999 | 0.998 | **1.000** | **1.000** | 0.999 | **1.000** | **1.000** |
| Right bundle-branch block | **1.000** | **1.000** | 0.999 | 0.999 | **1.000** | 0.999 | 0.999 | 0.999 | **1.000** | 0.999 |
| 2:1 AV block | **1.000** | 0.997 | **1.000** | 0.991 | 0.999 | 0.999 | 0.999 | 0.998 | 0.999 | 0.994 |
| Atrial fibrillation | **1.000** | **1.000** | **1.000** | 0.999 | 0.997 | 0.999 | **1.000** | 0.999 | 0.999 | 0.998 |
| Prolonged QT interval | 0.990 | 0.989 | 0.977 | 0.950 | 0.986 | 0.955 | 0.996 | 0.979 | **0.997** | 0.992 |
| Anterior MI | 0.995 | 0.995 | 0.996 | 0.991 | 0.986 | 0.998 | **0.999** | 0.994 | 0.995 | 0.981 |
| Sinus bradycardia | **0.995** | **0.995** | 0.994 | 0.989 | 0.991 | 0.994 | **0.995** | **0.995** | **0.995** | 0.994 |
| Early repolarization | 0.980 | 0.983 | 0.987 | **0.997** | 0.964 | 0.929 | 0.988 | 0.982 | 0.995 | 0.982 |

Table 15: Finetuning with a linear prediction head performance for the 6 labels, sorted by supervised S4 AUROC on the CODE-15% dataset.

| Label | ECGFounder | ECG-JEPA | ECGFM-KED | MERL | ST-MEM | HuBERT-ECG | ECG-FM | ECG-CPC | S4 | Net1D |
|---|---|---|---|---|---|---|---|---|---|---|
| Left bundle branch block | 0.997 | **0.998** | 0.997 | 0.994 | 0.994 | 0.996 | 0.994 | 0.997 | 0.996 | 0.995 |
| Sinus tachycardia | 0.994 | 0.994 | 0.994 | 0.992 | 0.988 | 0.993 | 0.991 | 0.994 | **0.995** | **0.995** |
| Right bundle branch block | 0.969 | 0.986 | 0.955 | **0.991** | 0.951 | **0.991** | 0.979 | 0.984 | 0.990 | 0.977 |
| Atrial fibrillation | **0.992** | **0.992** | 0.988 | **0.992** | 0.958 | 0.991 | 0.981 | **0.992** | 0.989 | 0.982 |
| 1st degree atrioventricular block | 0.986 | 0.985 | 0.984 | 0.967 | 0.959 | **0.987** | 0.980 | 0.984 | 0.984 | 0.976 |
| Sinus bradycardia | 0.985 | **0.994** | 0.929 | 0.958 | 0.935 | 0.990 | 0.990 | 0.980 | 0.984 | 0.969 |

Table 16: Finetuning with a linear prediction head performance for the 10 best-predicted labels, sorted by supervised S4 AUROC on the PTB-XL (all) dataset.

| Label | ECGFounder | ECG-JEPA | ECGFM-KED | MERL | ST-MEM | HuBERT-ECG | ECG-FM | ECG-CPC | S4 | Net1D |
|---|---|---|---|---|---|---|---|---|---|---|
| Subendocardial injury in inferior leads | 0.997 | 0.998 | **1.000** | 0.999 | 0.994 | 0.993 | 0.992 | 0.998 | **1.000** | 0.996 |
| Complete right bundle branch block | **0.999** | 0.998 | 0.998 | 0.998 | 0.997 | 0.997 | 0.998 | 0.998 | 0.998 | 0.997 |
| Complete left bundle branch block | 0.998 | 0.998 | 0.997 | 0.998 | 0.990 | 0.993 | **0.999** | 0.997 | 0.998 | 0.998 |
| Paroxysmal supraventricular tachycardia | **1.000** | 0.998 | 0.997 | 0.998 | 0.993 | 0.998 | **1.000** | 0.998 | 0.998 | 0.995 |
| Sinus tachycardia | **0.996** | 0.994 | 0.991 | 0.991 | 0.987 | 0.986 | 0.990 | **0.996** | 0.995 | 0.994 |
| Septal hypertrophy | 0.976 | 0.999 | 0.972 | 0.994 | 0.982 | 0.996 | 0.976 | 0.999 | 0.994 | **1.000** |
| Posterior myocardial infarction | **0.996** | 0.981 | 0.981 | 0.942 | 0.948 | 0.940 | 0.938 | 0.987 | 0.991 | 0.994 |
| Third degree AV block | **0.999** | 0.996 | 0.989 | 0.992 | 0.989 | 0.962 | 0.998 | 0.992 | 0.990 | 0.973 |
| Subendocardial injury in anteroseptal leads | 0.992 | **0.994** | 0.984 | 0.991 | 0.963 | 0.989 | 0.983 | 0.993 | 0.990 | 0.987 |
| Ventricular premature complex | 0.988 | 0.991 | 0.991 | 0.954 | 0.875 | **0.992** | 0.984 | 0.986 | 0.988 | 0.975 |

resentations between S0 and S4, which suggests that these layers learn quite similar representation,

Table 17: Finetuning with a linear prediction head performance for the 10 best-predicted labels, sorted by supervised S4 AUROC on the ZZU pECG dataset.

| Label | ECGFounder | ECG-JEPA | ECGFM-KED | MERL | ST-MEM | HuBERT-ECG | ECG-FM | ECG-CPC | S4 | Net1D |
|---|---|---|---|---|---|---|---|---|---|---|
| Ventricular escape complex(es) | **1.000** | 0.999 | **1.000** | 0.995 | 0.999 | 0.999 | **1.000** | **1.000** | 0.999 | 0.999 |
| Ventricular tachycardia | 0.997 | **0.999** | **0.999** | **0.999** | **0.999** | 0.994 | 0.998 | **0.999** | **0.999** | 0.998 |
| Sinus pause or arrest | 0.963 | 0.968 | 0.956 | 0.989 | 0.892 | 0.988 | 0.872 | 0.975 | **0.998** | 0.992 |
| AV block, advanced (high-grade) | 0.989 | **0.998** | 0.987 | 0.989 | **0.998** | 0.996 | 0.985 | 0.985 | **0.998** | 0.983 |
| Second-degree AV block | 0.999 | 0.999 | 0.999 | 0.971 | 0.978 | 0.958 | **1.000** | 0.994 | 0.998 | 0.974 |
| Right bundle-branch block | 0.995 | **0.997** | 0.995 | 0.982 | 0.988 | 0.988 | 0.992 | 0.995 | 0.989 | 0.990 |
| AV block, complete (third-degree) | 0.996 | **0.999** | **0.999** | 0.997 | 0.987 | 0.996 | 0.998 | 0.997 | 0.987 | 0.998 |
| Atrial flutter | 0.989 | **0.993** | 0.968 | 0.894 | 0.967 | 0.972 | 0.959 | 0.968 | 0.987 | 0.979 |
| TU fusion | 0.981 | 0.957 | 0.960 | **0.997** | 0.958 | 0.912 | 0.947 | 0.989 | 0.984 | 0.921 |
| Sinus bradycardia | 0.984 | 0.974 | 0.981 | 0.982 | 0.979 | 0.984 | 0.976 | **0.986** | 0.984 | 0.985 |

Table 18: Finetuning with a linear prediction head performance for the 10 best-predicted labels, sorted by supervised S4 AUROC on the EchoNext dataset.

| Label | ECGFounder | ECG-JEPA | ECGFM-KED | MERL | ST-MEM | HuBERT-ECG | ECG-FM | ECG-CPC | S4 | Net1D |
|---|---|---|---|---|---|---|---|---|---|---|
| lvef lte 45 | 0.901 | 0.897 | 0.896 | 0.902 | 0.883 | 0.878 | 0.873 | **0.910** | 0.897 | 0.879 |
| rv systolic dysfunction moderate or greater | 0.879 | 0.891 | 0.883 | 0.883 | 0.877 | 0.861 | 0.856 | **0.899** | 0.890 | 0.880 |
| aortic stenosis moderate or greater | 0.821 | 0.838 | 0.830 | 0.843 | 0.814 | 0.790 | 0.767 | 0.849 | **0.857** | 0.808 |
| tricuspid regurgitation moderate or greater | 0.832 | 0.845 | 0.840 | 0.845 | 0.825 | 0.806 | 0.779 | **0.851** | 0.844 | 0.827 |
| mitral regurgitation moderate or greater | **0.836** | 0.817 | 0.828 | 0.835 | 0.813 | 0.805 | 0.789 | 0.833 | 0.829 | 0.807 |
| pulmonary regurgitation moderate or greater | 0.875 | 0.857 | **0.880** | 0.872 | 0.853 | 0.825 | 0.861 | 0.870 | 0.828 | 0.851 |
| pasp gte 45 | 0.788 | 0.794 | 0.774 | 0.789 | 0.773 | 0.759 | 0.725 | **0.810** | 0.793 | 0.766 |
| tr max gte 32 | 0.779 | 0.792 | 0.757 | 0.781 | 0.764 | 0.748 | 0.722 | **0.804** | 0.784 | 0.748 |
| aortic regurgitation moderate or greater | 0.751 | 0.734 | 0.751 | 0.760 | 0.747 | 0.755 | 0.724 | 0.758 | **0.776** | 0.752 |
| lvwt gte 13 | 0.767 | 0.768 | 0.762 | 0.774 | 0.754 | 0.753 | 0.706 | **0.779** | 0.774 | 0.755 |

Table 19: Finetuning with a linear prediction head performance for the 10 best-predicted labels, sorted by supervised S4 AUROC on the MIMIC (cardiac) dataset.

| Label | ECGFounder | ECG-JEPA | ECGFM-KED | MERL | ST-MEM | HuBERT-ECG | ECG-FM | ECG-CPC | S4 | Net1D |
|---|---|---|---|---|---|---|---|---|---|---|
| I447 Left bundle block | 0.938 | 0.935 | 0.928 | 0.923 | 0.932 | 0.921 | 0.921 | 0.929 | **0.939** | 0.920 |
| I4510 Right bundle block | 0.936 | 0.925 | 0.931 | **0.937** | 0.930 | 0.912 | 0.915 | 0.919 | 0.918 | 0.926 |
| I5023 Acute/chronic systolic HF | 0.900 | 0.900 | 0.899 | 0.900 | 0.894 | 0.891 | 0.892 | **0.907** | 0.906 | 0.894 |
| I428 Other cardiomyopathies | 0.881 | 0.892 | 0.884 | 0.885 | 0.883 | 0.861 | 0.882 | 0.894 | **0.901** | 0.878 |
| I255 Ischemic cardiomyopathy | **0.902** | 0.882 | 0.896 | 0.895 | 0.888 | 0.879 | 0.870 | 0.892 | 0.900 | 0.873 |
| I132 HTN heart+CKD w/HF, ESRD | 0.897 | 0.899 | 0.873 | 0.893 | **0.913** | 0.847 | 0.886 | 0.912 | 0.896 | 0.876 |
| I482 Chronic AF | 0.895 | 0.894 | 0.892 | 0.888 | 0.862 | 0.865 | 0.882 | **0.899** | 0.893 | 0.872 |
| I211 STEMI, inferior wall | **0.896** | 0.881 | 0.867 | 0.876 | 0.838 | 0.776 | 0.794 | 0.885 | 0.887 | 0.754 |
| I078 Rheumatic tricuspid disease | 0.869 | 0.867 | **0.879** | 0.867 | 0.877 | 0.817 | 0.842 | 0.873 | 0.878 | 0.873 |
| I44 AV + LBBB | 0.876 | 0.869 | 0.876 | 0.866 | 0.868 | 0.855 | 0.869 | **0.879** | 0.875 | 0.850 |

Table 20: Finetuning with a linear prediction head performance for the 10 best predicted-labels, sorted by supervised S4 AUROC on the MIMIC (non-cardiac) dataset.

| Label | ECGFounder | ECG-JEPA | ECGFM-KED | MERL | ST-MEM | HuBERT-ECG | ECG-FM | ECG-CPC | S4 | Net1D |
|---|---|---|---|---|---|---|---|---|---|---|
| L9740 Chronic ulcer heel/midfoot | 0.841 | 0.864 | 0.762 | **0.908** | 0.897 | 0.764 | 0.886 | 0.904 | 0.941 | 0.805 |
| Z4502 ICD defibrillator management | 0.945 | 0.918 | 0.938 | 0.929 | 0.910 | **0.950** | 0.919 | 0.935 | 0.936 | 0.919 |
| K767 Hepatorenal syndrome | 0.909 | 0.886 | 0.812 | 0.885 | 0.813 | 0.631 | 0.856 | **0.924** | 0.903 | 0.801 |
| Z681 BMI 19.9 or less, adult | 0.885 | 0.896 | 0.861 | 0.896 | 0.881 | 0.812 | 0.890 | **0.918** | 0.901 | 0.824 |
| V850 Driver, construction vehicle accident | 0.850 | 0.871 | 0.850 | 0.891 | 0.912 | 0.711 | 0.877 | **0.913** | 0.893 | 0.810 |
| Z992 Renal dialysis dependence | 0.878 | 0.886 | 0.876 | 0.888 | 0.867 | 0.823 | 0.886 | **0.891** | 0.886 | 0.850 |
| V433 Car occupant, collision nontraffic | 0.863 | 0.894 | 0.885 | 0.881 | 0.858 | 0.805 | 0.890 | **0.915** | 0.885 | 0.871 |
| E660 Obesity, excess calories | 0.860 | 0.855 | 0.810 | 0.868 | 0.874 | 0.757 | 0.886 | **0.897** | 0.882 | 0.804 |
| N186 End-stage renal disease | 0.875 | 0.882 | 0.866 | 0.884 | 0.867 | 0.819 | 0.883 | **0.890** | 0.882 | 0.849 |
| V422 Outside car, motorcycle collision | 0.868 | **0.897** | 0.873 | 0.886 | 0.855 | 0.809 | 0.894 | 0.892 | 0.882 | 0.869 |

Table 21: Finetuning with a linear prediction head performance for the clinical deterioration labels, sorted by supervised S4 AUROC on the MIMIC (deterioration) dataset.

| Label | ECGFounder | ECG-JEPA | ECGFM-KED | MERL | ST-MEM | HuBERT-ECG | ECG-FM | ECG-CPC | S4 | Net1D |
|---|---|---|---|---|---|---|---|---|---|---|
| cardiac arrest | 0.868 | 0.861 | 0.871 | 0.882 | 0.852 | 0.825 | **0.895** | 0.887 | 0.858 | 0.801 |
| vasopressors | 0.771 | 0.768 | 0.764 | 0.760 | 0.739 | 0.719 | 0.771 | **0.800** | **0.800** | 0.774 |
| ecmo | 0.724 | 0.798 | 0.759 | 0.748 | 0.712 | 0.586 | **0.841** | 0.819 | 0.788 | 0.783 |
| mechanical ventilation | 0.786 | 0.777 | 0.784 | 0.781 | 0.765 | 0.741 | 0.785 | **0.802** | 0.785 | 0.759 |
| inotropes | 0.654 | 0.727 | 0.646 | 0.716 | 0.711 | 0.642 | 0.721 | 0.720 | **0.776** | 0.706 |
| severe hypoxemia | 0.498 | 0.553 | 0.458 | 0.573 | **0.591** | 0.470 | **0.591** | 0.555 | 0.530 | 0.559 |

suggesting a potential overparameterization in the middle of the network. The final two layers (S5, S6) show less similarity with earlier layer, suggesting that these are used for task specialization.

Table 22: Finetuning with a linear prediction head performance for the mortality labels, sorted by supervised S4 AUROC on the MIMIC (mortality) dataset.

| Label | ECGFounder | ECG-JEPA | ECGFM-KED | MERL | ST-MEM | HuBERT-ECG | ECG-FM | ECG-CPC | S4 | Net1D |
|---|---|---|---|---|---|---|---|---|---|---|
| mortality 1d | 0.829 | 0.837 | **0.895** | 0.843 | 0.807 | 0.825 | 0.849 | 0.876 | 0.835 | 0.794 |
| mortality 7d | 0.816 | 0.809 | 0.815 | 0.813 | 0.801 | 0.760 | 0.826 | **0.831** | 0.815 | 0.790 |
| mortality 28d | 0.811 | 0.811 | 0.795 | 0.808 | 0.791 | 0.739 | 0.814 | **0.823** | 0.807 | 0.780 |
| mortality 365d | 0.798 | 0.798 | 0.776 | 0.800 | 0.791 | 0.731 | 0.805 | **0.809** | 0.802 | 0.770 |
| mortality 90d | 0.791 | 0.795 | 0.770 | 0.791 | 0.779 | 0.719 | 0.800 | **0.803** | 0.796 | 0.765 |
| mortality 180d | 0.792 | 0.791 | 0.766 | 0.791 | 0.781 | 0.723 | 0.796 | **0.802** | 0.795 | 0.763 |
| mortality stay | **0.836** | 0.699 | 0.671 | 0.755 | 0.626 | 0.552 | 0.787 | 0.679 | 0.705 | 0.548 |

Table 23: Finetuning with a linear prediction head performance for the ICU labels, sorted by supervised S4 AUROC on the MIMIC (icu) dataset.

| Label | ECGFounder | ECG-JEPA | ECGFM-KED | MERL | ST-MEM | HuBERT-ECG | ECG-FM | ECG-CPC | S4 | Net1D |
|---|---|---|---|---|---|---|---|---|---|---|
| icu stay | 0.751 | 0.747 | 0.741 | 0.749 | 0.738 | 0.713 | 0.755 | **0.757** | 0.748 | 0.726 |
| icu 24h | 0.744 | 0.738 | 0.734 | 0.739 | 0.729 | 0.706 | 0.745 | **0.749** | 0.742 | 0.716 |

Table 24: Finetuning with a linear prediction head performance for the sex label, sorted by supervised S4 AUROC on the MIMIC (sex) dataset.

| Label | ECGFounder | ECG-JEPA | ECGFM-KED | MERL | ST-MEM | HuBERT-ECG | ECG-FM | ECG-CPC | S4 | Net1D |
|---|---|---|---|---|---|---|---|---|---|---|
| sex | 0.913 | 0.904 | 0.883 | 0.916 | 0.903 | 0.810 | 0.919 | **0.933** | 0.919 | 0.869 |

Table 25: Finetuning with a linear prediction head performance for the age label based on MAE on the MIMIC (age) dataset.

| Label | Units | ECGFounder | ECG-JEPA | ECGFM-KED | MERL | ST-MEM | HuBERT-ECG | ECG-FM | ECG-CPC | S4 | Net1D | Baseline |
|---|---|---|---|---|---|---|---|---|---|---|---|---|
| Age | Years | 8.78 | 8.82 | 8.88 | 8.54 | 9.59 | 11.03 | **7.835** | 8.32 | 8.66 | 9.86 | 15.07 |

Table 26: Finetuning with a linear prediction head performance for the biometrics labels, sorted by supervised S4 MAE on the MIMIC (biometrics) dataset.

| Label | Units | ECGFounder | ECG-JEPA | ECGFM-KED | MERL | ST-MEM | HuBERT-ECG | ECG-FM | ECG-CPC | S4 | Net1D | Baseline |
|---|---|---|---|---|---|---|---|---|---|---|---|---|
| Height | Inches | 2.69 | 2.70 | 2.71 | 2.66 | 2.78 | 3.02 | 2.804 | **2.58** | 2.63 | 2.86 | 3.30 |
| Weight | lbs | 27.85 | 28.27 | 28.34 | 27.33 | 29.79 | 32.04 | 26.417 | **26.38** | 27.53 | 30.16 | 35.73 |
| BMI | $kg/m^2$ | 4.14 | 4.12 | 4.23 | 4.05 | 4.40 | 4.67 | 3.943 | **3.90** | 4.06 | 4.43 | 5.30 |

Table 27: Finetuning with a linear prediction head performance for the ECG features labels, sorted by supervised S4 MAE on the MIMIC (ECG features) dataset

| Label | Units | ECGFounder | ECG-JEPA | ECGFM-KED | MERL | ST-MEM | HuBERT-ECG | ECG-FM | ECG-CPC | S4 | Net1D | Baseline |
|---|---|---|---|---|---|---|---|---|---|---|---|---|
| RR | ms | 107.01 | 106.59 | 106.78 | 108.28 | 108.98 | 108.93 | 112.913 | **105.73** | 105.94 | 109.96 | 154.38 |
| QRS | ms | 7.32 | 7.59 | 7.65 | 7.27 | 7.80 | 8.79 | 7.190 | 7.18 | **7.13** | 8.22 | 15.48 |
| QT | ms | 26.33 | 26.51 | 26.85 | 26.63 | 26.86 | 28.58 | 27.153 | **26.14** | 26.16 | 27.36 | 37.84 |
| QTc | ms | 18.09 | 17.89 | 18.16 | 18.36 | 17.91 | 20.67 | 17.657 | **17.41** | 17.60 | 19.48 | 26.84 |
| P wave axis | ° | 16.38 | 16.40 | 16.55 | 16.21 | 18.44 | 20.03 | 16.200 | **15.81** | 15.97 | 17.46 | 22.47 |
| QRS axis | ° | **15.07** | 15.41 | 16.01 | 15.86 | 18.22 | 15.849 | 15.35 | 15.35 | 16.41 | 36.06 | |
| T wave axis | ° | 24.36 | 24.19 | 24.75 | 24.59 | 28.31 | 30.95 | 24.648 | 24.05 | **23.94** | 25.34 | 35.41 |
| PT | sec | 2.84 | 2.84 | 2.94 | 2.86 | 2.92 | 3.04 | 2.824 | **2.80** | 2.82 | 2.93 | 3.85 |

**ECG-JEPA** Figure 4c shows the CKA analysis for the ECG-JEPA transformer model. Again the embedding layer shows little similarity to all other layers. The first transformer layer (Blk0) serves a transition point between the embedding layer and the following transformer layers, showing moderate similarity to both. As a striking feature, the following transformer layers (Blk1-Blk10) shows a very high similarity, which suggests that the representational structure barely changes across the 10 layers. This might be indicative for a representation collapse and possible inefficiency in the architecture where most layer contribute only minimally distinct representations. The representations of the final transformer layer (Blk11) deviate strongly from the previous, suggesting that the former is used for task specialization.

Table 28: Finetuning with a linear prediction head performance for the laboratory values predictions, sorted by supervised S4 MAE on the MIMIC (laboratory values) dataset

| Label | Units | ECGFounder | ECG-JEPA | ECGFM-KED | MERL | ST-MEM | HuBERT-ECG | ECG-FM | ECG-CPC | S4 | Net1D | Baseline |
|---|---|---|---|---|---|---|---|---|---|---|---|---|
| Albumin | g/dL | 0.44 | **0.43** | 0.44 | **0.43** | 0.45 | 0.49 | 0.442 | 0.44 | 0.44 | 0.44 | 0.49 |
| Anion Gap | mEq/L | 2.35 | 2.32 | 2.34 | 2.32 | 2.38 | 2.39 | 2.330 | 2.32 | **2.31** | 2.34 | 2.39 |
| Bicarbonate | mEq/L | 2.66 | 2.65 | 2.67 | 2.68 | 2.69 | 2.72 | 2.993 | **2.64** | 2.67 | 2.68 | 2.71 |
| Bilirubin, Total | mg/dL | **0.55** | 0.56 | 0.56 | **0.55** | **0.55** | 0.56 | 0.551 | 0.56 | **0.55** | **0.55** | 0.67 |
| Calcium, Total | mg/dL | **0.49** | **0.49** | **0.49** | 0.50 | 0.50 | 0.51 | 0.547 | 0.50 | **0.49** | **0.49** | 0.51 |
| Creatinine | mg/dL | 0.39 | 0.39 | 0.40 | 0.39 | 0.40 | 0.40 | 0.399 | 0.39 | **0.38** | 0.39 | 0.48 |
| Ferritin | ng/mL | 325.91 | 214.00 | 239.76 | 232.05 | 548.12 | 582.04 | 826.690 | 308.41 | **206.29** | 239.27 | 297.79 |
| Urea Nitrogen | mg/dL | 8.15 | 8.11 | 8.25 | 8.08 | 8.30 | 8.45 | 8.415 | **7.98** | 8.05 | 8.33 | 9.94 |
| Hematocrit | % | 3.88 | 3.90 | 3.96 | **3.87** | 3.93 | 4.12 | 3.901 | **3.87** | 3.88 | 4.01 | 4.34 |
| Hemoglobin | g/dL | 1.39 | 1.39 | 1.42 | 1.38 | 1.41 | 1.48 | 1.442 | **1.38** | 1.39 | 1.44 | 1.59 |
| Lymphocytes | % | 8.55 | 8.57 | 8.63 | 8.57 | 8.70 | 8.89 | **8.450** | 8.48 | 8.56 | 8.78 | 9.37 |
| MCHC | % | 1.10 | 1.08 | 1.09 | 1.08 | 1.10 | 1.10 | 1.082 | **1.07** | 1.07 | 1.08 | 1.13 |
| RDW | % | 1.22 | 1.23 | 1.23 | 1.22 | 1.23 | 1.28 | 1.218 | **1.21** | 1.21 | 1.24 | 1.42 |
| Red Blood Cells | m/μL | 0.47 | 0.46 | 0.47 | 0.46 | 0.47 | 0.49 | **0.456** | 0.46 | 0.46 | 0.48 | 0.52 |
| RDW-SD | fL | 4.48 | 4.47 | 4.52 | 4.50 | 4.58 | 4.71 | **4.465** | 4.55 | 4.49 | 4.61 | 5.31 |
| Creatine Kinase | IU/L | 264.66 | 232.67 | 229.90 | 237.43 | 272.06 | 249.10 | 238.532 | 243.92 | 242.04 | **225.86** | 275.63 |
| NTproBNP | pg/mL | 3769.34 | 3729.16 | 4037.84 | 3766.55 | 3699.87 | 3730.16 | 3683.700 | 3593.63 | 3655.23 | **3564.98** | 4538.09 |

Table 29: Finetuning with a linear prediction head performance for the vital signs predictions, sorted by supervised S4 MAE on the MIMIC (vital signs) dataset

| Label | Units | ECGFounder | ECG-JEPA | ECGFM-KED | MERL | ST-MEM | HuBERT-ECG | ECG-FM | ECG-CPC | S4 | Net1D | Baseline |
|---|---|---|---|---|---|---|---|---|---|---|---|---|
| dbp | mmHg | 11.41 | 11.38 | 11.43 | 11.41 | 11.48 | 11.65 | **11.353** | 11.36 | 11.41 | 11.58 | 11.83 |
| heartrate | bpm | 11.61 | 11.67 | 11.68 | 11.68 | 11.72 | 11.81 | 11.788 | **11.57** | **11.57** | 11.74 | 15.00 |
| O$_2$ sat | % | 1.54 | 1.54 | 1.55 | **1.53** | 1.55 | 1.57 | 1.535 | **1.53** | 1.54 | 1.56 | 1.64 |
| resprate | bpm | 2.20 | **2.18** | 2.19 | **2.18** | 2.25 | 2.20 | 2.232 | **2.18** | **2.18** | 2.19 | 2.29 |
| sbp | mmHg | 17.26 | 17.32 | 17.31 | 17.21 | 17.81 | 17.98 | **17.102** | 17.16 | 17.27 | 17.65 | 18.36 |
| temperature | $^\circ$F | 0.62 | **0.61** | **0.61** | 0.62 | 0.63 | 0.62 | 0.615 | **0.61** | **0.61** | 0.62 | 0.63 |

**ST-MEM** In Figure 4d, we see a similar overall structure as for the also transformer-based ECG-JEPA model. The embedding layer and the first transformer layer are distinct from a largely homogeneous block of following layers (Blk1-Blk9), the final layers (Blk10-Blk11) specialize. Compared to ECG-JEPA, ST-MEM shows a slightly better gradient through the middle layers, still the middle part of the architecture remains quite homogeneous, which again suggests wasted capacity. A remarkable difference compared to ECG-JEPA is the low similarity of the first transformer block (Blk0) with the following blocks (Blk1 onwards). This suggests a representational bottleneck early in the network which prevents the propagation of information from the embeddings into the following transformer blocks.

A.9.2 INTER-MODEL ANALYSIS

In Figure 5, we analyze the representational similarity across different models at three different points in the model architectures (early: after the encoder/stem/tokenizer; mid: after the first half of the processing layers; late: after the final processing layer).

In the early representations, shown in Figure 5a, the two transformers show high mutual similarity while the two non-transformer models show other patterns. This might be taken as a hint it is the architecture which shapes early representations and not the training objective. ECGFounder's early features are most dissimilar to all other features, indicating that the pure CNN learns qualitatively different early features.

Table 30: Fit parameters for the scaling analysis

| Model | $C$ | $\alpha$ | $L_0$ | $R^2$ |
|---|---|---|---|---|
| ECGFounder (pretrained) | 0.462 | 0.109 | 0.018 | 0.933 |
| ECGFounder (from scratch) | 0.887 | 0.270 | 0.120 | 0.998 |
| ECG-JEPA (pretrained) | 0.402 | 0.083 | $1.32 \times 10^{-13}$ | 0.989 |
| ECG-CPC (pretrained) | 0.463 | 0.104 | $4.35 \times 10^{-7}$ | 0.946 |
| ECG-CPC (scratch) | 0.501 | 0.101 | $9.13 \times 10^{-10}$ | 0.957 |
| S4 | 0.677 | 0.206 | 0.089 | 0.983 |

Table 31: Label efficiency for different training datasets

| Model | $r(N = 250)$ | $r(N = 500)$ | $r(N = 1000)$ | $r(N = 2000)$ |
|---|---|---|---|---|
| ECGFounder | 0.30 | 0.40 | 0.51 | 0.62 |
| ECG-JEPA | 0.11 | 0.17 | 0.27 | 0.40 |
| ECG-CPC | 0.21 | 0.27 | 0.34 | 0.40 |

Table 32: Unified comparison of computational cost, memory usage, and inference efficiency for all ECG representation learning models. Reported metrics include: (i) GFLOPs for forward (F) and backward (B) passes (lower is better), measured with batch size 1 on an NVIDIA L40; (ii) peak GPU memory during inference (lower is better), measured on PTB-XL (all) with batch size 64; and (iii) throughput (samples/s; higher is better) and latency (ms/sample; lower is better) under the same hardware and batch size. Timesteps correspond to the input sequence length for each model, and parameter counts include all trainable weights. The three best, i.e., most efficient, FMs are marked in bold face. The overall best model is additionally underlined.

| Model | Timesteps | Parameters ↓ | GFLOP (F/B) ↓ | GPU Mem (MB) ↓ | Thr↑/ Lat↓ |
|---|---|---|---|---|---|
| ECGFounder (CNN) | 1250 | 33.8M | **0.602 / 5.066** | **187.37** | **2220.67 / 0.450** |
| ECG-JEPA (Transformer) | 2500 | 87.2M | 73.877 / 221.6 | 2136.79 | 98.71 / 10.131 |
| ST-MEM (Transformer) | 600 | 90.3M | 20.926 / 62.779 | 609.71 | 368.64 / 2.713 |
| MERL (CNN) | 1250 | **4.6M** | **0.863** / 2.582 | **86.24** | **2727.36 / 0.367** |
| ECGFM-KED (CNN) | 5000 | **9.7M** | 5.423 / 16.258 | **427.39** | **1631.28 / 0.613** |
| HuBERT-ECG (Transformer) | 500 | 97.2M | 18.829 / 69.779 | 845.27 | 213.87 / 4.676 |
| ECG-FM (Transformer) | 2500 | 93.9M | 29.242 / 109.794 | 1091.71 | 692.35 / 1.444 |
| ECG-CPC (SSM) | 600 | **3.8M** | **1.741 / 5.213** | 482.32 | 442.38 / 2.260 |

Table 33: Impact of layer-dependent learning rates on model performance. AUROC scores comparing models trained with and without layer-dependent learning rate scheduling. Layer-dependent learning rates apply different learning rates across network layers, typically with lower rates for earlier layers. Higher AUROC values indicate better performance. Best scores are marked in bold face. Underlined values indicate models pretrained on the corresponding dataset.

| Model | PTB-XL (all) With ↑ / Without ↑ | EchoNext With ↑ / Without ↑ | CPSC-Extra With ↑ / Without ↑ |
|---|---|---|---|
| ECGFounder (CNN) | 0.934 / **0.936** | 0.817 / **0.818** | 0.906 / **0.907** |
| ECG-JEPA (Transformer) | **0.940** / 0.779 | **0.817** / 0.730 | **0.897** / 0.790 |
| ST-MEM (Transformer) | **0.908** / 0.891 | **0.816** / 0.751 | **0.883** / 0.853 |
| MERL (CNN) | **0.925** / 0.919 | 0.822 / **0.823** | **0.873** / 0.852 |
| ECGFM-KED (CNN) | 0.889 / **0.918** | 0.806 / **0.817** | 0.824 / **0.833** |
| HuBERT-ECG (Transformer) | **0.915** / 0.499 | **0.792** / 0.500 | **0.876** / 0.513 |
| ECG-FM (Transformer) | **0.927** / 0.504 | **0.772** / 0.499 | **0.862** / 0.517 |
| ECG-CPC (SSM) | **0.949** / 0.938 | **0.831** / 0.828 | **0.898** / 0.875 |

Table 34: Comparison of aggregated macro-AUROC scores for different model architectures trained from scratch. Higher AUROC values indicate better performance. The best-performing result is highlighted in boldface and underlined, while models that do not perform statistically significantly worse are also highlighted in boldface.

| Model | PTB-XL (all) ↑ | EchoNext ↑ | CPSC-Extra ↑ |
|---|---|---|---|
| S4 (SSM) | **0.941** | **0.819** | 0.852 |
| Net1D (CNN) | 0.929 | **0.818** | 0.803 |
| Mamba-1 (SSM) | 0.895 | 0.770 | **0.858** |
| Mamba-2 (SSM) | 0.914 | 0.807 | 0.844 |

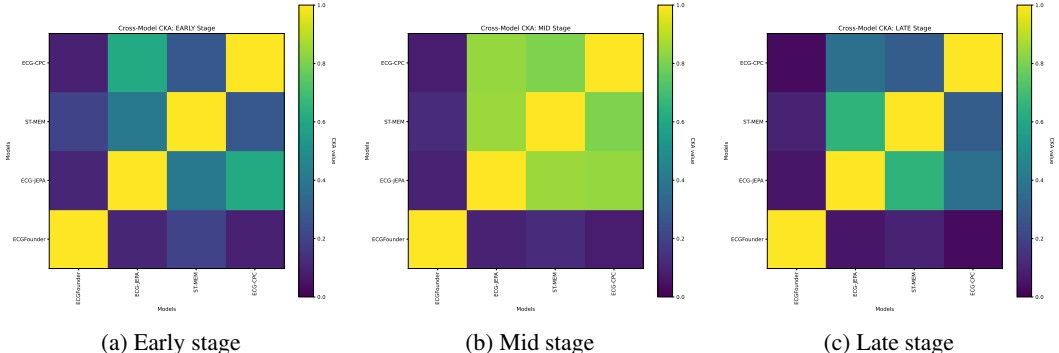

(a) Early stage        (b) Mid stage        (c) Late stage

Figure 5: Inter-model representational similarity across network depths. CKA heatmaps comparing corresponding stages across four FMs (ECGFounder, ECG-JEPA, ST-MEM, ECG-CPC). Higher values (yellow) indicate similar representations between layers. CKA computed using Gaussian RBF kernel ($\sigma = 1.0$) on 2,500 samples of PTB-XL (all) dataset per model.

This result should be compared to the CKA comparison of the late features, which is shown in Figure 5c. The two transformer models (ECG-JEPA and ST-MEM) remain highly similar, whereas ECGFounder remains very dissimilar to all three other models. Also the dissimilarity of ECG-CPC and the transformer models increases in later layers, which supports the hypothesis that the ECG-CPC model learns distinct representations and does not replicate the patterns learned by the transformer models.

### A.9.3 INSIGHTS

To summarize, in the hybrid CNN-SSM model (ECG-CPC) the CNN encoder extracts local features while the S4 layers progressively build temporal abstractions. Each component serves a distinct purpose, and you see representational change at each layer. The CNN-based ECGFounder shows smooth gradients, but still high similarities across blocks S0-S4 indicate redundancies, which could possibly alleviated by decreasing the number of layers. Both considered transformer models show nearly identical representations in the middle 8-10 layers, which puts into question the efficiency of the archtitecture. It raises the question why transformer models show this particular pattern on ECG data specifically. The isolated Blk0 in the case of ST-MEM indicate poor integration of encoder and further processing layers, which might explain the deficiencies in downstream performance observed for this model.

The inter model comparisons reveal strong similarities between the two transformer models in spite of different training objectives. The results indicate potential for ensembling of ECGFounder with any of the other models to maximize diversity.

The CKA analysis provided hints for ST-MEM's weakness (early bottleneck) and ECG-CPC's strengths (efficient, purposeful layer utilization), which establishes CKA as a useful diagnostic tool for ECG FMs.

### A.10 AUROC VS. SPECIFICITY VS. $F_1$-SCORE

In this section, we shed light on alternative performance metrics beyond macro AUROC. The clinically most commonly used performance metrics such as sensitivity and specificity require the specification of a decision threshold. While this decision should be taken based on clinical desiderata in a disease-specific manner, we restrict for simplicity to a disease-unspecific threshold. More specifically, we adjust the decision threshold for every condition such a sensitivity of 0.8 is reached. We then assess model performance according to macro-averaged specificity and $F_1$-score. Due to its widespread use in the literature, we base the experiment on the PTB-XL on the most finegrained level of the label hierarchy covering 71 unique labels. We compare the performance of foundation models after finetuning. The results are compiled in Figure 6.

As a non-trivial finding, the model ranking according to AUROC completely agrees with the model ranking according to specificity, as it suggests that better overall discriminative power according to

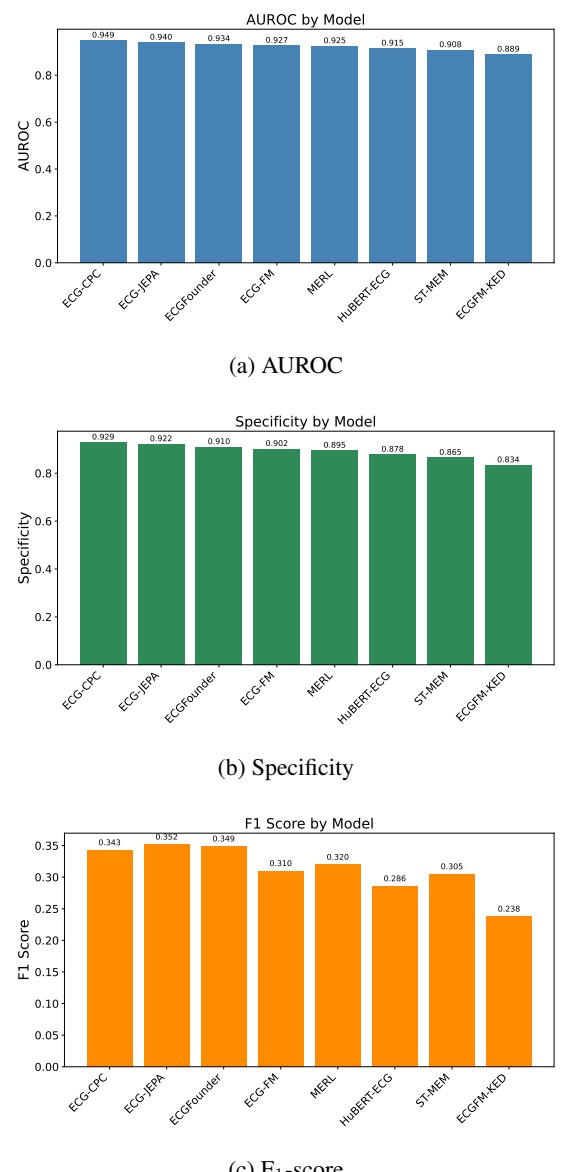

(a) AUROC

(b) Specificity

(c) F$_1$-score

Figure 6: Performance comparison of foundation models on PTB-XL under the finetuning with linear head evaluation mode. Evaluated using (a) AUROC, (b) Specificity, and (c) F$_1$-score at target sensitivity of 0.8. Higher values indicate better performance.

AUROC translates into better model performance in terms of clinically relevant performance metrics sensitivity/specificity at relevant operating points. The F$_1$-score assesses only precision/positive predictive value as the recall/sensitivity was fixed to 0.8 for all models. The model ranking based on F$_1$-score shows correlation with the ranking based on AUROC but does not fully agree. We see the agreement in terms of specificity as the more crucial observation as the positive predictive value depends on the prevalence of the respective conditions in the dataset, which, however, does not follow a proper population-level distribution.

