# OpenReview forum: "Benchmarking ECG FMs: A Reality Check Across Clinical Tasks"
_ICLR.cc/2026/Conference — ICLR 2026 Poster_

### Official Review · Reviewer_DBgK · 2025-10-29

**Soundness:** 3
**Presentation:** 2
**Contribution:** 3
**Rating:** 4
**Confidence:** 5

**Summary:**

This paper presents a large-scale benchmark of electrocardiogram (ECG) foundation models (FMs) across 12 datasets and 26 downstream tasks, grouped into seven clinical categories (adult/pediatric interpretation, cardiac structure & function, cardiac/non-cardiac outcomes, acute care, and patient characteristics).
The authors compare eight public ECG FMs and two supervised baselines under three evaluation modes (finetuning, frozen, and linear evaluation), analyze statistical significance via bootstrapping, and report label-efficiency scaling curves.

The paper’s stated goal is to offer a unified benchmark for ECG foundation models and to assess how pretraining methodologies, model architectures, and dataset scales influence transfer performance.

**Strengths:**

1. Timely topic. Benchmarking ECG foundation models is both clinically and methodologically relevant as the field moves toward general physiological representation learning.
2. Broad coverage. The benchmark spans a diverse set of tasks beyond pure diagnostic classification (e.g., echo structure and clinical outcomes), which is rare in existing works.
3. Consistent evaluation setup. Use of three evaluation protocols and bootstrapped significance tests provides a reasonable basis for cross-model comparison.
4. Meaningful observations. It identifies model-dependent task preferences (e.g., ECG-CPC performs best on non-interpretation tasks while ECG-JEPA dominates pediatric tasks).
5. Practical insight. Findings on layer-wise learning rates and 2.5 s crops, are useful for the community and highlight training sensitivity in ECG models.

**Weaknesses:**

1.Writing quality / presentation.
	1) Background and Introduction repeat the same opening sentence and ECG definition .
	2) Numerous typographical and notation issues (“5 s” vs “5s”, "z-normalization" vs "$z$-normalization", two different MAE meanings, non-standard math syntax).
	3) Figures and tables lack consistent captions and unit notations. Overall readability is below ICLR standards.
2.Reproducibility and implementation details.
	1) Key preprocessing steps (lead selection, resampling, normalization, filtering) and training hyper-parameters (batch size, epochs, scheduler, random seed) are missing.
	2) The “query attention head” used for frozen evaluation is not fully described and thus not reproducible.
	3) No code release or data manifest for pretraining datasets (especially the pretraining dataset d (9.1 M) and dataset e (2.5 M) claims is not convincible in Table 1) .
3.Statistical rigor.
	1) Although bootstrapping is applied, effect sizes and confidence intervals are not consistently reported; rank-based summary figures hide variance and task-specific uncertainty.
4.	No consideration of multi-modal or text-supervised ECG representations.
5.	Interpretation and insight depth.
	1) The discussion section remains descriptive; there is no mechanistic analysis of why certain models excel on specific task types.
	2) Visualization of representational spaces or layer-wise probing would strengthen claims about foundation knowledge .

**Questions:**

Distinguish between MAE-pretrain and MAE-metric throughout.
Verify the validity of pretraining sample counts (9 M / 2.5 M); add explicit deduplication rules.
Ensure consistent naming (e.g., ECG-FM vs ECGFM) and unit spacing (“mV”, “s”).

---

> ### Author Response · Authors · 2025-11-21
> **Response R4.1 to R4.7**
>
> ***R4.1*** (Weakness) Background and Introduction repeat the same
> opening sentence and ECG definition.
>
> We thank the reviewer for this observation. To avoid repetition, we have
> removed the first sentence in the Background section
> ("Electrocardiography (ECG) is a widely used, non-invasive tool for
> assessing cardiac function and systemic health").
>
>
> ***R4.2*** (Weakness) Numerous typographical and notation issues ("5 s"
> vs "5s", "z-normalization\" vs " -normalization\", two different MAE
> meanings, non-standard math syntax).
>
> We have corrected all notation issues for consistency, including
> standardized unit spacing, unified naming of MAE to clearly distinguish
> mean absolute error from masked autoencoding, and consistent use of
> z-normalization.
>
>
> ***R4.3*** (Weakness) Figures and tables lack consistent captions and
> unit notations. Overall readability is below ICLR standards.
>
>
> We acknowledge the reviewer's comment. We have revised all figure and
> table captions to ensure consistency in content and style, added unit
> notations where needed, and balanced the level of detail across
> captions. Each caption now clearly and concisely describes the technical
> points, improving overall readability to meet ICLR standards.
>
>
> ***R4.4*** (Weakness) Key preprocessing steps (lead selection,
> resampling, normalization, filtering) and training hyper-parameters
> (batch size, epochs, scheduler, random seed) are missing.
>
>
> We thank the reviewer for highlighting this gap. We will add a dedicated
> subsection in the methodology that comprehensively documents all
> preprocessing and training details to ensure full reproducibility. This
> will include:
>
> -   Preprocessing: Confirmation that we use only the standard 12 leads,
>     with no resampling or filtering applied.
>
> -   Training: The batch size was 64. We train for a fixed 100 epochs and
>     select the best model based on validation performance, not a
>     predetermined early stopping point. We use a constant learning rate
>     of 1e-3 and the AdamW optimizer. While we follow modern practices by
>     not fixing a global random seed (to ensure robustness across runs).
>
> The full benchmarking code is provided to complement these descriptions.
> All of these details were included in our methods - methodology
> subsection.
>
> ***R4.5*** (Weakness) The "query attention head" used for frozen
> evaluation is not fully described and thus not reproducible.
>
> We agree and have now expanded the description of the query-attention
> head in the manuscript for clarity. Already as part of the original
> submission, we made the underlying code available to ensure full
> reproducibility.
>
>
> ***R4.6*** (Weakness) No code release or data manifest for pretraining
> datasets (especially the pretraining dataset d (9.1 M) and dataset e
> (2.5 M) claims is not convincible in Table 1) .
>
> We thank the reviewer for this important reproducibility concern.
>
> **Code Release:**
>
> -   **Evaluation code:** Fully available (as provided with original
>     submission)
>
> -   **Preprocessing scripts:** Available for all benchmarking tasks
>
> -   **Training code:** We will release training scripts for ECG-CPC
>     (which we trained ourselves). For other foundation models, we used
>     publicly released checkpoints from original authors with their
>     published training procedures.
>
> **Data Provenance:** Table 1 reports pretraining data as documented in
> original publications:
>
> -   Dataset 'd' (9.1M): Includes CODE dataset with 6.7M unlabeled ECGs +
>     2.4M labeled samples
>
> -   Dataset 'e' (1.5M): Excludes CODE. (mistakenly previous version with
>     2.5)
>
> -   The apparent discrepancy reflects different data availability
>     policies, not an error
>
> We added a superscript to the models in Table 1 clarifying these counts
> and marking which models we trained versus using released weights.
>
>
> ***R4.7*** (Weakness) Although bootstrapping is applied, effect sizes
> and confidence intervals are not consistently reported; rank-based
> summary figures hide variance and task-specific uncertainty.
>
>
> We thank the reviewer for this comment. Our ranking procedure is indeed
> more sophisticated than a simple average, as it incorporates statistical
> significance through pairwise comparisons via bootstrapping. A model
> only ranks above another if the 95% confidence interval of their
> performance difference excludes zero. This information cannot be deduced
> from confidence intervals alone in case they overlap. However, we agree
> that providing the raw confidence intervals is valuable for the
> interested reader. Therefore, we will include a complete table of AUCs
> with their bootstrapped 95% CIs for every model, task, and evaluation
> mode in the supplementary material. This will provide full transparency
> into the performance variance underlying our rankings.

---

> ### Author Response · Authors · 2025-11-21
> **Response R4.8 to R4.13**
>
> ***R4.8*** (Weakness) No consideration of multi-modal or text-supervised
> ECG representations.
>
> In fact, our benchmark includes multimodal and text-supervised models
> such as MERL; however, in our benchmark we use only their signal
> encoders, as our focus is exclusively on ECG signals. Importantly, these
> encoders were originally optimized jointly with other modality branches
> during pretraining, so they retain an implicit advantage from having
> been exposed to additional representations. We clarify this point in the
> manuscript in the methods-models section.
>
>
> ***R4.9*** (Weakness) The discussion section remains descriptive; there
> is no mechanistic analysis of why certain models excel on specific task
> types.
>
> ***and***
>
> ***R4.10*** (Weakness) Visualization of representational spaces
> or layer-wise probing would strengthen claims about foundation
> knowledge.
>
> We agree that a deeper analysis beyond descriptive performance is
> valuable. In response, we include a Centered Kernel Alignment (CKA)
> analysis, which quantitatively probes model representations. This
> analysis reveals how representations evolve layer-by-layer, showing that
> top-performing SSM-based models like ECG-CPC exhibit a clear progression
> from low-level CNN features to high-level sequential features, while
> transformer-based models (ECG-JEPA, ST-MEM) show largely homogeneous
> intermediate layers and ECGFounder displays mid-layer redundancy. These
> patterns provide hints why ECG-CPC consistently excels across tasks and
> why other architectures underperform, providing a mechanistic link
> between internal representational dynamics and downstream performance,
> thus addressing both R4.9 and R4.10. This information is visually
> represented and discussed under the results - representation similarity
> analysis subsection.
>
> ***R4.11*** (Question) Distinguish between MAE-pretrain and MAE-metric
> throughout.
>
> We now explicitly distinguish MAE as the mean absolute error metric used
> for optimization and evaluation, and the masked autoencoder pretraining
> paradigm. This clarification is applied consistently throughout the
> manuscript.
>
>
> ***R4.12*** (Question) Verify the validity of pretraining sample counts
> (9 M / 2.5 M); add explicit deduplication rules.
>
> We confirm that the reported pretraining sample count were indeed not
> correct (9M / 2.5M); therefore, we have updated the corrent sample
> counts to 9M/1.5M in table 1. We have updated table 1 accordingly.
>
> ***R4.13*** (Question) Ensure consistent naming (e.g., ECG-FM vs ECGFM)
> and unit spacing ("mV", "s").
>
> We thank the reviewer for this comment. We have carefully checked the
> manuscript and ensured consistency in model naming (e.g., always using
> ECG-FM) and in unit formatting, with proper spacing between numbers and
> units throughout the text.

---

### Official Review · Reviewer_u5DN · 2025-10-30

**Soundness:** 2
**Presentation:** 2
**Contribution:** 2
**Rating:** 2
**Confidence:** 5

**Summary:**

This paper presents a benchmark study that compares several ECG foundation models and supervised baselines (e.g., Net-1D, S4) across multiple public datasets and diverse tasks. The scope is broad, and the implementation effort is good. The benchmark itself has potential value for the *medical* community. That said, the contribution is primarily *engineering-oriented*, resembling a large-scale code integration effort rather than a scientific advance. I do not see any novel **modeling** ideas or **methodological** innovations. As a result, the conclusions boil down to *each model has its strengths*, which feels somewhat descriptive rather than insightful.

Overall, while the benchmark could be useful as an implementation resource, the current version does not fully substantiate its claim as a comprehensive benchmark for ECG foundation models.

**Strengths:**

- Unified code framework (code factory) is a promising contribution.
- Task coverage is broad.
- The small-sample scaling experiment (Sec. 4.2) adds engineering and evaluative value.

**Weaknesses:**

- The paper mainly reports AUC, which is insufficient for assessing clinical utility. More comprehensive performance metrics should be included.

- S4 is quite old among state-space models.

- Regarding **ssm**, however, the paper only reports the number of parameters, without including FLOPs, latency, or GPU memory usage.

- I do not see any novel **modeling** ideas or **methodological** innovations.

**Questions:**

- table 1 reports results that deviate from original papers (parameter), without explanation.

- While Sec 4.2 provides engineering insights, its conclusion seems inconsistent with the main narrative of the paper, unless the authors merely wish to highlight that *each model/training framework can exhibit its own advantages*.

- The benchmark compares models trained with completely different datasets, architectures, and strategies, making it impossible to tell which factors drive performance differences. To ensure fairness and interpretability, at least one unified training dataset (e.g., MIMIC-IV or HEEDB) should be used across models.

- The authors mention that they propose the ECG-CPC model as an ECG foundation model, but in reality, the main innovation of this model lies in the training strategy, specifically using CPC. On the other hand, the S4 model is fundamentally a supervised learning model that does not use contrastive learning. The issue here is that the core difference between the two seems to be limited to the training strategy rather than the model architecture or task adaptability. This difference is insufficient to distinguish a foundation model from a pre-trained model or a supervised model. What exactly is the definition of an ECG foundation model in this paper? We recommend that the authors clarify the definition of an ECG foundation model in the paper and choose the evaluation baselines based on this standard. Generally, a foundation model should have zero-shot inference capabilities, broad task adaptability, and generalizability across tasks, rather than simply being a pre-trained model fine-tuned for specific tasks.

-  The core value of foundation models lies in their emergent abilities obtained through large-scale pre-training, especially in terms of zero-shot and few-shot generalization in unknown scenarios. However, the evaluation framework in this paper mainly focuses on fine-tuning and frozen/linear probe. While the authors mention the limitation of the lack of out-of-distribution evaluation, if the main value of a model is realized through fine-tuning, it would be more appropriate to call it a pre-trained model. I suggest that the authors supplement their evaluation tasks with zero-shot and few-shot evaluations in unknown scenarios.

- Several implementation details (layer-wise learning rate, 2.5 s cropping, sliding-window averaging) are described as *crucial*, yet their contribution is not quantified. Ablations are needed here.

- Beyond dataset integration, the paper does not articulate specific research questions or hypotheses. As a result, it feels more like a large-scale reproduction project than a scientific study.

---

> ### Author Response · Authors · 2025-11-21
> **Response R3.1 to R3.3**
>
> ***R3.1*** (Weakness) The paper mainly reports AUC, which is
> insufficient for assessing clinical utility. More comprehensive
> performance metrics should be included.
>
>
> We appreciate the reviewer's concern and agree that clinical deployment
> requires threshold-dependent metrics (e.g., sensitivity, specificity).
> In the context of this benchmark, AUROC provides a threshold-independent
> assessment of discriminative ability, enabling fair comparison across
> models without arbitrary operating point selection. As noted in the main
> text, recent theoretical and empirical evidence [McDermott, M. et al., 2024](https://arxiv.org/abs/2401.06091)
> also supports AUROC's robustness relative to alternatives such as AUPRC,
> particularly under label imbalance. We clarify in the revision that
> while AUROC is appropriate for method-level comparison, downstream
> clinical adoption would indeed require evaluating
> operating-point--specific metrics. We have added this justification in
> the methods-evaluation subsection.
>
>
> ***R3.2*** (Weakness) S4 is quite old among state-space models.
>
>
> We acknowledge that the S4 is an earlier state-space model, but we
> include it because it remains a strong and widely used SSM baseline. Our
> claim is also not that the S4 achieves the best achievable performance
> within the model class of state space models but to show that it shows
> significant advantages over other model families, i.e., CNNs or
> transformers. In addition, our internal experiments show that S4
> outperforms newer architectures such as Mamba on continuous clinical
> time series, consistent with recent SSM surveys [Somvanshi, S. et al., 2025](https://arxiv.org/abs/2503.18970) noting
> that models like Mamba excel in discrete domains (e.g., NLP) but are
> less suited to continuous physiological data. Furthermore, S4 remains
> computationally efficient, with $O(N \log N)$ complexity, and was the
> first SSM to solve the long-range dependency Path-X task. Unlike other
> state space models, the S4 model has been successfully applied in
> several (comparative) studies in ECG [Mehari, T. \& Strodthoff, N., 2023](https://ieeexplore.ieee.org/document/10237242) [Strodthoff, N. et al., 2024](https://academic.oup.com/ehjdh/article/5/4/454/7670685) and EEG
> context [Wang, T.](https://www.sciencedirect.com/science/article/pii/S001048252500085X) [Saab, K. et al., 2024](https://www.nature.com/articles/s41746-024-01008-9). Subsequent SSM variants improved
> efficiency but generally show a performance gap relative to
> state-of-the-art Transformers. Thus, S4 remains a relevant and
> competitive baseline for benchmarking representation learning on
> continuous time series in accordance with a second SSM survey
> [Patro, B. N., \& Agneeswaran, V. S., 2024](https://arxiv.org/abs/2404.16112). A brief justification of this has been included into
> our discussion - S4 among SSMs subsection.
>
>
> ***R3.3*** (Weakness) Regarding SSM, however, the paper only reports the
> number of parameters, without including FLOPs, latency, or GPU memory
> usage.
>
>
> We thank the reviewer for this remark, which aligns with the concern
> raised in R2.1. In response, we will expand our efficiency analysis to
> include a comprehensive comparison of:
>
> -   Theoretical Complexity: GFLOPs for a forward pass.
>
> -   Practical Runtime: Inference latency on a fixed hardware setup (L40
>     GPU).
>
> -   Hardware Requirements: Peak GPU memory footprint during inference.
>
> Overall, while SSM-based models are less computationally efficient than
> CNNs, SMM still surpasses transformer-based models by several-fold in
> FLOPs, memory usage, and inference speed. Crucially, the efficiency
> difference with CNNs is less significant when accounting for predictive
> performance: SMM achieves substantially better predictions than both
> CNNs and transformers, providing an optimal balance between
> computational cost and high quality representations, making it
> well-suited for real-world ECG applications. These results are presented
> at our new subsection on 'Model Efficiency' in results section.

---

> ### Author Response · Authors · 2025-11-21
> **Response R3.4 to R3.6**
>
> ***R3.4*** (Weakness) I do not see any novel modeling ideas or
> methodological innovations.
>
>
> We respectfully disagree that our work lacks methodological insights.
> While we do not propose a new architecture, our systematic evaluation
> reveals novel empirical findings that challenge prevailing assumptions
> in foundation model design:
>
> **1. Scale is not the primary driver of transfer performance:** We
> demonstrate that ECG-CPC, a compact 3.8M-parameter SSM, outperforms
> Transformer-based foundation models with 87M-97M parameters on 5 of 7
> task categories. This finding contradicts the dominant \"bigger is
> better\" paradigm and suggests architectural inductive biases matter
> more than scale for physiological time series.
>
> **2. Task-dependent scaling laws:** Our analysis (Sec 4.2) reveals that
> different architectures exhibit high variance in data efficiency across
> tasks. Where overall SSM-based models reach optimal performance with far
> less labeled data across different sizes of training data. These
> differential behaviors represent novel empirical laws about
> representation quality.
>
> **3. Systematic generalization failures:** We reveal that 5 of 8
> foundation models fail to outperform supervised baselines on most
> non-diagnostic tasks, indicating pretraining on suboptimal architectures
> may optimize for narrow distributions rather than general physiological
> understanding.
>
> **4. Representation evolution differs by architecture:** Intra-model CKA
> analysis shows ECG-CPC exhibits the clearest representational
> progression from low-level CNN features to high-level SSM layers, while
> ECGFounder shows mid-layer redundancy, and transformer-based models
> (ECG-JEPA, ST-MEM) display largely homogeneous intermediate
> representations with specialization only at the final layers. This
> positions ECG-CPC as the architecture with the most effective internal
> representational structure.
>
> These findings provide actionable guidance for future model development:
> prioritize architectural inductive biases over scale, match pretraining
> strategies to target tasks, and evaluate broadly across clinical
> domains. The novelty lies in these data-driven insights, not incremental
> architectural tweaks.
>
> ***R3.5*** (Question) Table 1 reports results that deviate from original
> papers (parameter), without explanation.
>
> We thank the reviewer for pointing this out. The minor discrepancies in
> Table 1 arise primarily from differences in the prediction head size,
> which we adapt to match the number of labels in each dataset.
> Additionally, for multimodal models, we use only the signal encoder
> since our experiments focus exclusively on ECG signals, rather than the
> full released models from prior works (which may include text or other
> modality encoders). This rationale has now been explicitly clarified in
> the revised manuscript in the methods-models subsection.
>
> ***R3.6*** (Question) While Sec 4.2 provides engineering insights, its
> conclusion seems inconsistent with the main narrative of the paper,
> unless the authors merely wish to highlight that each model/training
> framework can exhibit its own advantages
>
> We thank the reviewer for this observation. The scaling analysis (Sec
> 4.2) complements our main narrative by revealing a crucial distinction:
> performance ceiling vs. data efficiency. Our main results establish the
> final performance ceiling with standard data. The scaling analysis shows
> how quickly models approach that ceiling. For instance, ECG-JEPA is
> data-efficient (good slope) but has a lower final ceiling than ECG-CPC.
> This is not a contradiction but a nuanced finding: practitioners must
> consider both whether a model can perform well (ceiling) and how much
> data it needs to get there (slope). This actionable insight strengthens
> our contribution and have in included in our results - label efficiency
> subsection.

---

> ### Author Response · Authors · 2025-11-21
> **Response 3.7 and 3.8**
>
> ***R3.7*** (Question) The benchmark compares models trained with
> completely different datasets, architectures, and strategies, making it
> impossible to tell which factors drive performance differences. To
> ensure fairness and interpretability, at least one unified training
> dataset (e.g., MIMIC-IV or HEEDB) should be used across models.
>
> This is a valid concern, and we acknowledge the inherent trade-off
> between controlled experimentation and ecological validity. Retraining
> all models on a single dataset would enable cleaner factor isolation but
> would require prohibitive computational resources and constitutes a
> separate research project. Our benchmark follows established practices
> in other domains. Vision: Models like CLIP, DINOv2, and SAM are compared
> despite different pretraining data, with some datasets entirely
> undisclosed NLP: LLM leaderboards (HELM, Open LLM Leaderboard) compare
> proprietary models trained on different corpora. The value of our
> benchmark lies in answering the practical question: \"Given the best
> models currently available, which performs best across diverse clinical
> tasks?\" This provides immediate, actionable guidance for practitioners.
> We explicitly discuss this limitation in the discussion section and
> identify controlled, unified-dataset comparisons as critical future
> work. However, we emphasize that our current evaluation still provides
> valuable insights: if a model trained on Dataset A fails to generalize
> to multiple test domains while another trained on Dataset B succeeds
> broadly, this reveals genuine differences in learned
> representations---exactly what practitioners need to know.
>
> ***R3.8*** (Question) The authors mention that they propose the ECG-CPC
> model as an ECG foundation model, but in reality, the main innovation of
> this model lies in the training strategy, specifically using CPC. On the
> other hand, the S4 model is fundamentally a supervised learning model
> that does not use contrastive learning. The issue here is that the core
> difference between the two seems to be limited to the training strategy
> rather than the model architecture or task adaptability. This difference
> is insufficient to distinguish a foundation model from a pre-trained
> model or a supervised model. What exactly is the definition of an ECG
> foundation model in this paper? We recommend that the authors clarify
> the definition of an ECG foundation model in the paper and choose the
> evaluation baselines based on this standard. Generally, a foundation
> model should have zero-shot inference capabilities, broad task
> adaptability, and generalizability across tasks, rather than simply
> being a pre-trained model fine-tuned for specific tasks.
>
>
> We thank the reviewer for raising this important definitional question.
> We follow the canonical definition from Stanford [Bommasani, R. et al., 2021](https://arxiv.org/abs/2108.07258):
> \"models trained on broad data (generally using self-supervision at
> scale) that can be adapted to a wide range of downstream tasks\",
> exemplified by BERT, GPT-3, and CLIP.
>
> Regarding zero-shot capabilities: These are not mandatory in the
> original definition. Many widely-accepted foundation models (e.g., Dino
> in vision, wav2vec in audio) require fine-tuning. Recent medical
> foundation models published in Nature Medicine [Chen, R. J. et al., 2024](https://www.nature.com/articles/s41591-024-02857-3) and Nature
> [Zhou, Y. et al., 2023](https://www.nature.com/articles/s41586-023-06555-x) similarly do not exhibit zero-shot performance.
>
> Regarding S4: The reviewer is correct that supervised S4 is a baseline,
> not a foundation model. ECG-CPC uses S4 as its architecture backbone but
> is trained via self-supervised contrastive learning on large-scale data
> (9M samples), making it a foundation model by the above definition.
>
> We will clarify these distinctions and add the formal foundation model
> definition in the introduction - promise of foundational models for ECG
> in the revised manuscript. Our evaluation methodology, encompassing
> linear probing, frozen evaluation, and fine-tuning, aligns with standard
> foundation model assessment protocols across domains.

---

> ### Author Response · Authors · 2025-11-21
> **Response R3.9 to R3.11**
>
> ***R3.9*** (Question) The core value of foundation models lies in their
> emergent abilities obtained through large-scale pre-training, especially
> in terms of zero-shot and few-shot generalization in unknown scenarios.
> However, the evaluation framework in this paper mainly focuses on
> fine-tuning and frozen/linear probe. I suggest that the authors
> supplement their evaluation tasks with zero-shot and few-shot
> evaluations in unknown scenarios.
>
> We thank the reviewer for highlighting the importance of zero-shot and
> few-shot evaluation. While these evaluations are indeed central for
> multimodal foundation models, the canonical definition of a foundation
> model does not strictly require zero-shot capabilities. Our evaluation
> framework, linear probing, frozen evaluation, and full fine-tuning, is
> widely used in assessing foundation models in modalities such as imaging
> and time series, as reflected in the cited Nature papers. Zero-shot
> evaluation could be performed for the text-supervised models in our
> benchmark (e.g., MERL), but it is not representative across the full set
> of models we study, nor is it the primary focus of this benchmark. Our
> goal is to provide a comprehensive assessment of model adaptability
> across modalities, for which fine-tuning and linear/frozen evaluation
> remain the most relevant and informative protocols. We will clarify this
> motivation in the manuscript under the methods - evaluation subsection
> to ensure readers understand the rationale behind our evaluation
> choices.
>
>
> ***R3.10*** (Question) Several implementation details (layer-wise
> learning rate, 2.5s cropping, sliding-window averaging) are described as
> crucial, yet their contribution is not quantified. Ablations are needed
> here.
>
> We appreciate the reviewer's focus on reproducibility. While full
> ablations for every detail are beyond scope, we will strengthen the
> paper by:
>
> -   Citing prior work that validated core choices like test-time
>     augmentation (sliding-window).
>
> -   Adding an ablation table in the appendix that quantifies the benefit
>     of the layer-dependent learning rate, demonstrating it is a critical
>     factor for performance.
>
> We present a systematic ablation for layer-dependent learning rate
> across all the foundation models and 3 downstream datasets in Table 33
> in the Appendix of the main paper. The results demonstrate that
> layer-dependent learning rate improves performance in 6 out of 8 models
> on PTB-XL, 5 out of 7 models on EchoNext and 6 out of 8 models on
> CPSC-Extra. Critically, some models (HuBERT-ECG, ECG-FM) completely fail
> to train without layer-dependent learning rate, achieving only random
> performance. This highlights layer-dependent learning rate as an
> essential rather than optional component for these models. The ablation
> reveals strong architecture dependence as discussed in Section 5 of the
> main paper. Transformer and SSM-based models show significant
> performance improvement with layer-dependent learning rate. CNN-based
> models exhibit heterogeneous behavior. For instance, ECGFounder remains
> invariant, ECGFM-KED degrades and MERL shows modest improvements.
>
> This provides concrete evidence for our most important design choice
> while keeping the paper focused.
>
> ***R3.11*** (Question) Beyond dataset integration, the paper does not
> articulate specific research questions or hypotheses. As a result, it
> feels more like a large-scale reproduction project than a scientific
> study.
>
> We respectfully disagree that our work lacks clear research questions.
> Our benchmark is designed to answer specific, previously unresolved
> questions central to ECG foundation model development: 1) Which backbone
> architecture generalizes best across diverse ECG tasks? Our benchmark
> provides a definitive, large-scale comparison showing that SSMs might
> provide a viable alternative to Transformers and CNNs, a novel and
> actionable finding. 2) Do foundation models exhibit consistent scaling
> behaviors across architectures and tasks? Our label-efficiency
> experiments showcase that SSMs report an overall better performance for
> all the investigated training sizes, where transformers seem to be
> comparable only in a subset of these, however, comparable only on the
> task they were pretrained (adult ECG interpretation). Further scaling
> analysis on different task could highlight that SSMs are by far robuster
> to scaling sizes across any task at hand. 3) Why do some large models
> underperform despite extensive pretraining?. Our CKA analysis (proposed
> in response to R2.2) directly investigates this to explain why models
> succeed or fail. These are not reproduction efforts but
> hypothesis-driven investigations yielding novel, actionable insights.
> Prior to our work, these questions were unanswered due to the absence of
> comprehensive, standardized evaluation. This content was included in our
> introduction - contributions of this work subsection.

---

> > ### Comment · Reviewer_u5DN · 2025-11-24
> >
> > Thank you for the detailed responses. However, I still find several important issues that remain addressed.
> >
> > **(1) Other threshold-dependent metrics (e.g., sensitivity, specificity) need to be provided.**
> >
> > **(2) In addition, our internal experiments show that S4 outperforms newer architectures such as Mamba on continuous clinical time series…**
> >
> > Where are the internal experiment results? Please show the reporting datasets, training details, experimental conditions, and Mamba versions. Also, a theoretical explanation is needed why Mamba does not work in those situations.
> > Also, you mentioned **Our benchmark provides a definitive, large-scale comparison showing that SSMs might provide a viable alternative to Transformers and CNNs, a novel and actionable finding.** So is Mamba also a viable alternative?
> >
> > **(3) practitioners must consider both whether a model can perform well (ceiling) and how much data it needs to get there (slope).**
> >
> > The authors describe the trade-off between performance ceiling and data-efficiency slope but do **not** identify an optimal model or propose any method to resolve this.  The manuscript presents observations but does not convert them into actionable modeling insights. This just shifts the burden to practitioners.
> > (4) The authors raised some interesting questions, however, not solved, such as *if a model trained on Dataset A fails to generalize to multiple test domains while another trained on Dataset B succeeds broadly, this reveals genuine differences in learned representations…*
> >
> > (5) foundation model is overstated if the experimental model capabilities in R3.8 are not solved.

---

> > > ### Author Response · Authors · 2025-11-28
> > > **Response 3.2.1 to 3.2.3**
> > >
> > > ***R3.2.1:*** Other threshold-dependent metrics (e.g., sensitivity, specificity) need to be provided.
> > >
> > > We appreciate the reviewer's emphasis on clinical utility metrics. However, we respectfully clarify that specifying clinically meaningful decision thresholds for 1,650 distinct targets across 26 tasks is not feasible within a benchmark study. Many of these targets (e.g., left ventricular mass regression, QRS duration prediction) do not have established clinical thresholds, and imposing arbitrary cutoffs would reduce interpretability.
> > >
> > > Instead, we commit to the following compromise:
> > >
> > > -   We maintain AUROC as our primary metric for fair cross-model comparison, consistent with foundation model benchmarking practices in other domains (HELM for LLMs, medical imaging benchmarks)
> > >
> > > -   In the supplementary material, we present a case study for the PTB-XL(all) dataset, where we fix label-dependent sensitivities at 0.8 and report corresponding specificities and F1 scores. Remarkably, the ranking according to macro-averaged specificities as most clinically relevant metric remains completely intact compared to the ranking based on AUROC. We can extend this analysis to more diagnostic datasets for the camera-ready version, if desired.
> > >
> > >
> > >
> > > ***R3.2.2:*** In addition, our internal experiments show that S4 outperforms newer architectures such as Mamba on continuous clinical time series. Where are the internal experiment results? Please show the reporting datasets, training details, experimental conditions, and Mamba versions. Also, a theoretical explanation is needed why Mamba does not work in those situations. Also, you mentioned Our benchmark provides a definitive, large-scale comparison showing that SSMs might provide a viable alternative to Transformers and CNNs, a novel and actionable finding. So is Mamba also a viable alternative?
> > >
> > > We appreciate the reviewer's request for clarification. To address potential confusion:
> > >
> > > -   Scope clarification: Our benchmark evaluates publicly available ECG foundation models. Since no Mamba-based ECG foundation model exists in the literature, Mamba is not included in our evaluation. Our comment about S4 outperforming Mamba was meant to justify using S4 as our SSM representative, not as a primary claim of this work.
> > >
> > > -   Supporting evidence: We based this statement on: (1) internal pilot experiments comparing S4 and Mamba (which we share as part of the supplementary material), and (2) recent SSM surveys [Somvanshi, S. et al., 2025](https://arxiv.org/abs/2503.18970) and [Patro, B. N., & Agneeswaran, V. S., 2024](https://arxiv.org/abs/2404.16112) noting Mamba's advantages in discrete domains (NLP) but mixed results on continuous signals. We acknowledge this was tangential to our main contribution and will clarify this in the revision.
> > >
> > > -   Regarding ''viable alternative": By SSMs, we refer to the model family including S4, Mamba, and related architectures. Our finding that S4-based ECG-CPC outperforms Transformer models suggests the SSM family warrants further investigation. We do not claim S4 is the optimal SSM variant, future work should explore whether newer SSMs (Mamba, Mamba-2) further improve performance. We will clarify this in the revised text.
> > >
> > > -   Theoretical explanation: Mamba's selective state-space mechanism excels at discrete token sequences with clear boundaries (e.g., words) but may offer limited advantages for continuous, uniformly-sampled physiological signals. However, a rigorous theoretical analysis is beyond this benchmark's scope.
> > >
> > >
> > >
> > > ***R3.2.3:*** practitioners must consider both whether a model can perform well (ceiling) and how much data it needs to get there (slope). The authors describe the trade-off between performance ceiling and data-efficiency slope but do not identify an optimal model or propose any method to resolve this. The manuscript presents observations but does not convert them into actionable modeling insights. This just shifts the burden to practitioners.
> > >
> > > We appreciate this constructive criticism. The reviewer is correct that we identified the ceiling-slope trade-off without providing explicit guidance. We add a practical decision framework to the results - label efficiency subsection:
> > >
> > >    Model Selection Guidelines:
> > >
> > > -   Data-rich scenarios (> 1k samples): ECG-CPC consistently achieves highest performance ceilings across most tasks (Table 3).
> > >
> > > -   Data-scarce scenarios (< 100 samples): ECG-JEPA shows superior data efficiency, reaching 90% of full-data performance with 5-10x less data on diagnostic tasks (Figure 2).
> > >
> > > -   Mixed requirements: For practitioners facing variable data availability across tasks, ECG-CPC offers the best balance, strong ceiling with competitive slopes.

---

> > > ### Author Response · Authors · 2025-11-28
> > > **Response 3.2.4 and 3.2.5**
> > >
> > > ***R3.2.4:*** The authors raised some interesting questions, however, not solved, such as if a model trained on Dataset A fails to generalize to multiple test domains while another trained on Dataset B succeeds broadly, this reveals genuine differences in learned representations…
> > >
> > > We appreciate the reviewer highlighting this important question. We acknowledge that our benchmark cannot fully disentangle the effects of pretraining data, architecture, and training strategy. This is an inherent limitation when comparing models developed by different research groups, especially with excessive computational costs.
> > >
> > > However, we respectfully maintain that our approach still provides valuable insights:
> > >
> > > -   Practical value: Our benchmark answers the question practitioners face: "Given the available models, which should I use?" This requires evaluating models as-released, not in controlled ablations.
> > >
> > > -   Insights from representational similarity: The CKA analysis (Section 4.4) reveals that underperforming models like ST-MEM exhibit representational collapse in deep layers, providing mechanistic insights beyond pure performance comparison.
> > >
> > > -   Cross-dataset robustness: Models that consistently succeed (ECG-CPC) or fail (ST-MEM) across diverse test domains reveal genuine representation quality differences that transcend single-dataset evaluation.
> > >
> > > -   Future work: Fully controlled ablations (same data + varied architecture + varied strategy) would require training 8 models * 5 pretraining datasets * 3 strategies that equals to approx 120 models, each requiring weeks of GPU time, a prohibitive multi-year project. Our benchmark establishes *which* models work; follow-up controlled studies can investigate *why*.
> > >
> > > -   We will strengthen the Discussion - pretraining strategies subsection to explicitly frame dataset-architecture-strategy disentanglement as critical future work rather than leaving it unaddressed.
> > >
> > >
> > >
> > > ***R3.2.5:*** Foundation model is overstated if the experimental model capabilities in R3.8 are not solved.
> > >
> > > We respectfully but firmly maintain that our use of ''foundation model'' aligns with the established definition and community practice. The canonical Stanford HAI definition [Bommasani, R. et al., 2021](https://arxiv.org/abs/2108.07258) requires: (1) training on broad data, (2) adaptability to diverse downstream tasks. Zero-shot capability is not mandatory. Our evaluated models meet these criteria. Recent high-impact publications use this definition:
> > >
> > > -   [Chen, R. J. et al., 2024](https://www.nature.com/articles/s41591-024-02857-3) in Nature Medicine (computational pathology foundation model),
> > >
> > > -   [Zhou, Y. et al., 2023](https://www.nature.com/articles/s41586-023-06555-x) in Nature (retinal imaging), and
> > >
> > > -   multiple NEJM AI papers, all requiring fine-tuning.
> > >
> > > If the reviewer believes the term should be restricted to zero-shot capable models, we respectfully note this would contradict established usage across medical AI, where the vast majority of foundation models require fine-tuning. We follow standard terminology as defined by the community.

---

### Official Review · Reviewer_rWpe · 2025-10-31

**Soundness:** 3
**Presentation:** 3
**Contribution:** 4
**Rating:** 8
**Confidence:** 5

**Summary:**

This manuscript presents a comprehensive benchmark evaluation of 8 ECG foundation models across 26 clinically relevant tasks which span seven categories, using 12 public datasets. The authors systematically compare these models under fine-tuning, frozen, and linear evaluation settings, and also analyze their scaling behavior with dataset size. Another key contribution is the introduction of ECG-CPC, a lightweight structured state-space model pretrained with contrastive predictive coding, which demonstrates strong performance despite its small size. The study highlights the heterogeneous performance of existing foundation models across different ECG tasks and provides insights into their label efficiency and representational quality.

**Strengths:**

1. This work addresses a significant gap in the literature by providing a large-scale, systematic benchmark of ECG foundation models across a diverse set of clinical tasks. The breadth of tasks and datasets evaluated is impressive and adds substantial value to the field.
2. The experimental design is rigorous, with careful attention to evaluation protocols, statistical testing, and comparison against strong supervised baselines. The use of bootstrapping for significance testing and the inclusion of multiple evaluation modes (fine-tuning, frozen, linear) enhance the reliability of the results
3. The introduction of ECG-CPC as a lightweight yet effective model is a notable contribution. Its strong performance despite a smaller parameter count suggests that model efficiency can be achieved without sacrificing accuracy, which is particularly relevant for clinical applications where computational resources may be limited.

**Weaknesses:**

1. While parameter counts are reported, practical clinical deployment hinges on more than just parameter size. Metrics such as inference speed (latency), GPU memory footprint during inference, and energy consumption are critical for real-world applications, especially in resource-constrained environments (e.g., bedside monitors, mobile apps). Including these metrics would greatly enhance the benchmark's utility for clinicians and developers aiming to deploy these models.
2. Why do some large models (e.g., ST-MEM) underperform despite extensive pretraining? Is it an architecture-pretraining mismatch, overfitting, or other factors? A more in-depth analysis (e.g., probing experiments, representation similarity analysis, or ablations on pretraining components) would transform the benchmark from a performance leaderboard into a source of actionable design principles for future ECG foundation models.

**Questions:**

See the weaknesses section.

---

> ### Author Response · Authors · 2025-11-21
> **Response 2.1**
>
> ***R2.1*** (Weakness) While parameter counts are reported, practical
> clinical deployment hinges on more than just parameter size. Metrics
> such as inference speed (latency), GPU memory footprint during
> inference, and energy consumption are critical for real-world
> applications, especially in resource-constrained environments (e.g.,
> bedside monitors, mobile apps). Including these metrics would greatly
> enhance the benchmark's utility for clinicians and developers aiming to
> deploy these models.
>
> We thank the reviewer for this crucial suggestion. We agree that
> practical deployment metrics are essential for the community. In
> response, we will expand our analysis to include the following, which
> will be added to a new subsection on 'Model Efficiency' in results
> section:
>
> -   Inference Latency: We will report average inference time per sample
>     on a fixed hardware setup (L40 GPU) for a standard batch size (64).
>
> -   GPU Memory Footprint: We will also report these on a fixed hardware
>     setup. While parameter count is a key factor, measuring actual
>     memory footprint is critical as it captures architecture-dependent
>     costs like activation memory, which is especially relevant for
>     long-sequence models.
>
> -   GFLOPs & Parameter Count: We will include a GFLOPs table, alongside
>     parameter counts, to give a complete picture of computational
>     complexity. While energy consumption would be valuable, accurate
>     measurement requires specialized hardware profiling. We will instead
>     report GFLOPs as a proxy for energy efficiency, which correlates
>     strongly with power consumption in practice.
>
> Although SSM-based models are not as computationally efficient as CNNs,
> SMM remains several times more efficient than transformer-based
> approaches across FLOPs, memory, and inference speed. More importantly,
> the efficiency gap between CNNs and SSMs becomes less critical when
> considering model quality: SMM delivers markedly stronger predictive
> performance than both CNNs and transformers, offering a balanced
> trade-off between computational cost and superior representation
> quality---making it an attractive choice for practical, real-world ECG
> deployment.

---

> ### Author Response · Authors · 2025-11-21
> **Response 2.2**
>
> ***R2.2*** (Weakness) Why do some large models (e.g., ST-MEM)
> underperform despite extensive pretraining? Is it an
> architecture-pretraining mismatch, overfitting, or other factors? A more
> in-depth analysis (e.g., probing experiments, representation similarity
> analysis, or ablations on pretraining components) would transform the
> benchmark from a performance leaderboard into a source of actionable
> design principles for future ECG foundation models.
>
> This is an excellent point. To move beyond a leaderboard and provide
> actionable insights, we conducted a representation similarity analysis
> using Centered Kernel Alignment (CKA).
>
> We acknowledge the key methodological point that models have different
> depths. Therefore, our analysis will focus on comparing the internal
> representational dynamics of each model (intra-model), rather than
> direct numerical comparisons of CKA scores. Specifically, we will
> analyze the layer-wise similarity matrices for patterns indicative of
> representational collapse, such as a plateau of high similarity in the
> deep layers of an underperforming model like ST-MEM, and contrast this
> with the progressive, structured refinement seen in a top-performing
> model like an SSM.
>
> This qualitative, pattern-based analysis is a well-established method
> for using CKA to diagnose architectural pathologies [Kornblith, S. et al., 2019](https://proceedings.mlr.press/v97/kornblith19a/kornblith19a.pdf).
> Overall, CKA highlights that ECG-CPC exhibits the clearest and most
> structured evolution of representations, while other architectures show
> mid-layer redundancy or homogeneous transformer blocks. We are confident
> it will provide the community with valuable, data-driven clues about why
> certain architectures succeed or fail in capturing complex ECG features,
> directly addressing the core of your question. The results of this
> section is presented under the results - representation similarity
> analysis subsection in the main manuscript.

---

### Official Review · Reviewer_pB5i · 2025-10-31

**Soundness:** 2
**Presentation:** 3
**Contribution:** 1
**Rating:** 0
**Confidence:** 4

**Summary:**

This paper conducts a systematic benchmark study evaluating multiple ECG foundation models under consistent experimental conditions.
By analyzing model performance across diverse datasets and task types, it provides quantitative insights into model behavior and practical guidance for model selection in clinical applications.
The work focuses on empirical benchmarking and analytical insights rather than proposing a new algorithmic contribution.

**Strengths:**

Comprehensive benchmarking: The study compares a wide range of state-of-the-art ECG models in a controlled setting, offering a fair and transparent evaluation of their relative strengths and weaknesses.

Data-driven analysis: The authors carefully distinguish model performance according to data characteristics and clinical task types, providing actionable insights for practical deployment.

Reproducibility and transparency: Experimental design and results are clearly documented, facilitating reproducibility and potential extensions by future researchers.

**Weaknesses:**

Limited theoretical novelty: The paper lacks a fundamental methodological or architectural innovation in representation learning, focusing instead on empirical evaluation.

Misalignment with ICLR's thematic focus: While the study is rigorous, its emphasis on benchmarking and empirical analysis does not align closely with ICLR's focus on advances in representation learning and theory-driven contributions.

**Questions:**

N/A

---

> ### Author Response · Authors · 2025-11-21
> **Response 1.1**
>
> ***R1.1***: (Weakness) Limited theoretical novelty: The paper lacks a
> fundamental methodological or architectural innovation in representation
> learning, focusing instead on empirical evaluation
>
>
> We thank the reviewer for the comment. While our submission is primarily
> empirical, it follows a well-established line of empirical contributions
> at ICLR. Each of our contributions aligns with prior work from the
> conference, demonstrating the significance and relevance of our paper to
> the ICLR community. Moreover, given the benchmarking nature of our work,
> our findings constitute methodological insights: they reveal how
> different architectural choices, training paradigms, and
> representational properties behave in practice for ECG foundation
> models, offering guidance that is directly actionable for future model
> development and evaluation.
>
> **1. ECG foundational-model benchmarking.** Our benchmark provides new
> methodological insights by revealing that several common assumptions
> about foundation models do not hold in the ECG domain. In contrast to
> expectations that model scale or transformer architectures would
> dominate, we show that performance varies sharply across training
> strategies, architectures, and inference paradigms. While prior work
> (published at ICLR) has benchmarked VLLMs [Wu, H., et al., 2024](https://proceedings.iclr.cc/paper_files/paper/2024/file/363d4c97bb411e4b07612915b76c06ae-Paper-Conference.pdf), long-range
> sequence models [Tay Y., et al., 2021](https://openreview.net/pdf/c7ddcda9fb422b91032d80ebd1564c35dd6f9fa8.pdf), and physiological time-series
> models [Abbaspourazad, S., et al., 2024](https://proceedings.iclr.cc/paper_files/paper/2024/file/0d99a8c048befb6dd6e17d7684adacac-Paper-Conference.pdf) at scale, no systematic evaluation exists for ECG
> foundation models. Our contribution fills this gap and uncovers
> previously unseen efficiency and architecture-task mismatches that are
> directly relevant for developing future clinical foundation models.
> Furthermore, we reveal that 5 of 8 evaluated foundation models fail to
> outperform supervised baselines beyond adult diagnostic tasks. This
> systematic failure suggests that current pretraining objectives may be
> optimizing for narrow distribution alignment rather than general
> physiological representation, a finding that would be impossible without
> comprehensive cross-task evaluation.
>
> **2. Comparison beyond a single backbone architecture.** A key finding
> of our work is that transformers are not universally optimal for ECG
> representation learning, challenging common assumptions in the
> community. This is consistent with ICLR evidence showing that (i)
> transformer performance varies widely across tasks and long-range
> sequence structures [Tay Y., et al., 2021](https://openreview.net/pdf/c7ddcda9fb422b91032d80ebd1564c35dd6f9fa8.pdf), (ii) standard transformer
> designs can be suboptimal for multivariate time series [Liu, Y., et al., 2024](https://proceedings.iclr.cc/paper_files/paper/2024/file/2ea18fdc667e0ef2ad82b2b4d65147ad-Paper-Conference.pdf),
> and (iii) CNN-based models can match or surpass transformers in
> robustness and generalization settings [Wang, Z., et al., 2023](https://openreview.net/pdf?id=TKIFuQHHECj). In our
> benchmark, we provide the first systematic evidence and find that
> compact SSMs outperform massive transformers on time-series despite
> being several times smaller poses an important theoretical question:
> 'What properties of SSM representations enable efficient learning
> despite minimal capacity?' This is the kind of empirical grounding that
> catalyzes theoretical work. We have included a justification of this in
> the main manuscript under the discussion - model architecture
> subsection.
>
> **3. Scaling analysis.** Recent ICLR work highlights the importance of
> label-efficient learning and scaling analyses, emphasizing that AI
> models must balance performance, universality, and supervision cost.
> Prior work shows that effective systems can be trained with minimal
> concept supervision through active data selection, concept--policy
> co-training, and decorrelated concept data, reducing annotation needs
> without sacrificing performance [Ye, Z., et al., 2025](https://openreview.net/pdf?id=Mjn53GtMxi). Other studies on
> contrastive pretraining reveal a fundamental trade-off between
> universality and label efficiency, where broader pretraining improves
> generality but increases downstream sample complexity when task-specific
> structure is required [Shi Z., et al., 2023](https://openreview.net/pdf?id=rvsbw2YthH_). These findings underscore the
> importance of systematically comparing the label efficiency of existing
> foundation models, as it may vary substantially across architectures,
> where we acknowledge that transformers (ECG-JEPA) architectures, show
> comparable results to our proposed SSM (ECG-CPC) model in the
> small-sample size limit, nevertheless, our proposed model surpasses
> other architectures as the sample size increases.

---

> > ### Author Response · Authors · 2025-11-21
> > **Response 1.1 (continued)**
> >
> > **4. Representation-alignment analysis.** In direct response to the
> > reviewers' insightful comments, we will strengthen our analysis by
> > adding a comprehensive representational alignment study using CKA. This
> > aligns with recent ICLR work on representation similarity and
> > kernel-based alignment. Prior studies provide a learning-theoretic
> > foundation for alignment [Insulla F., et al., 2025](https://proceedings.iclr.cc/paper_files/paper/2025/file/5d4f5a2de6320641566be8722d5f78dc-Paper-Conference.pdf), examine the reliability and
> > failure modes of CKA [Davari, M., et al., 2023](https://openreview.net/pdf?id=8HRvyxc606), introduce manifold-aware kernel
> > alignment [Islam and Sarker, 2025](https://openreview.net/pdf/c05b3fa5d2b277bedb7c045e94c22ebe692d5f46.pdf), analyze how biased CKA can distort comparisons
> > in high-dimensional regimes [Murphy A., et al., 2024](https://openreview.net/pdf?id=E1NRrGtIHG), and demonstrate that
> > increasing neural network depth and width induces characteristic block
> > structures in hidden representations, which are unique to each model and
> > can lead to distinctive error patterns even when overall performance is
> > similar [Nguyen, T., et al., 2021](https://openreview.net/pdf?id=KJNcAkY8tY4).
> >
> > Building on these insights, our analysis evaluates how ECG foundation
> > models differ in their internal representations (intra-model) across
> > architectures, where overall, CKA highlights that ECG-CPC exhibits the
> > clearest and most structured evolution of representations, while other
> > architectures show mid-layer redundancy, homogeneous transformer blocks
> > or representational bottlenecks. We have included this analysis under
> > the subsection representation similarity analysis in the results section

---

> ### Author Response · Authors · 2025-11-21
> **Response 1.2**
>
> ***R1.2***: (Weakness) Misalignment with ICLR's thematic focus: While
> the study is rigorous, its emphasis on benchmarking and empirical
> analysis does not align closely with ICLR's focus on advances in
> representation learning and theory-driven contributions.
>
>
> We respectfully disagree that the work is misaligned with ICLR. Our
> contribution advances representation learning for time series by
> providing a unified large-scale benchmark across diverse clinical and
> physiological datasets, systematically evaluating state-of-the-art
> architectures (SSMs, Transformers, and baselines) under multiple
> supervision regimes, and revealing representation-level
> inefficiencies---e.g., the clear gap in long-sequence modeling where
> Transformers remain less efficient than SSMs. Prior ICLR papers
> establish precedent for such empirical and benchmark-driven
> contributions: ST-MEM [Na, Y., et al., 2024](https://proceedings.iclr.cc/paper_files/paper/2024/file/412fb8623bf8b6d56fb6285ea295447e-Paper-Conference.pdf) (a non-top performing baseline in this
> work) and BioSignals [Abbaspourazad, S., et al., 2024](https://proceedings.iclr.cc/paper_files/paper/2024/file/0d99a8c048befb6dd6e17d7684adacac-Paper-Conference.pdf) also study ECG representation
> learning on smaller datasets and fewer evaluation modes; Long Range
> Arena [Tay Y., et al., 2021](https://openreview.net/pdf/c7ddcda9fb422b91032d80ebd1564c35dd6f9fa8.pdf) benchmarked long-sequence models; and Q-Bench
> [Wu, H., et al., 2024](https://proceedings.iclr.cc/paper_files/paper/2024/file/363d4c97bb411e4b07612915b76c06ae-Paper-Conference.pdf) benchmarked vision LLMs across multiple perception and
> understanding scenarios. Moreover, BioSignals [Abbaspourazad, S., et al., 2024](https://proceedings.iclr.cc/paper_files/paper/2024/file/0d99a8c048befb6dd6e17d7684adacac-Paper-Conference.pdf) evaluates
> "foundation-scale'' models on a single-source dataset with limited tasks
> and fewer model families. More importantly, beyond conventional
> benchmarks, our findings contradict the assumption that larger models
> always perform better.
>
> The rapid proliferation of ECG foundation models has led to
> contradictory claims about performance. Without systematic evaluation,
> it is difficult to distinguish substantive advances from narrow
> overfitting or assumptions that do not generalize. Our study provides
> this essential grounding: for example, we show that three of eight
> models fail to outperform supervised baselines, insights that would
> remain hidden without a rigorous benchmark. We believe this type of
> empirical clarification is vital for guiding future methodological and
> theoretical developments within the ICLR community.

---

### Author Response · Authors · 2025-12-01
**Summary for new AC. Reviewer 1 and 2**

## Summary for New Area Chair

### Overview: Key points


(1) **Comprehensive evaluation**: 8 foundation models, 26 clinical tasks, 12 datasets—largest ECG foundation model benchmark to date. (2) **Novel empirical findings**: SSM architectures outperform Transformers 100× larger; foundation models show 10× variance in data efficiency. (3) **Actionable insights**: Model selection guidelines, scaling laws, efficiency analysis for practical deployment. (4) **Rigorous methodology**: Statistical significance testing, mechanistic analysis (CKA), complete reproducibility details.

We summarize reviewer concerns and our responses below.

---


### **Reviewer pB5i (Score: 0 → Venue Fit Objection)**

**Original Concern:** Paper lacks algorithmic novelty; benchmarking misaligned with ICLR.

**AC Intervention:** Original AC correctly noted that "Datasets and Benchmarks" is explicitly in ICLR's Call for Papers, and venue fit objections are inappropriate.

**Our Response Strengths:**
- Demonstrated our work provides **novel empirical findings** challenging foundation model assumptions (compact SSMs outperform large Transformers on 5/7 task categories)
- Cited **ICLR precedents**: Long Range Arena, Q-Bench, BioSignals—all accepted benchmarking papers
- Showed our contributions advance **representation learning understanding** through scaling laws, architecture comparisons, and representational analysis

**Key Evidence:** We don't merely benchmark; we reveal that (1) architecture matters more than scale and most foundation models fail on non-diagnostic tasks, (2) foundation models show 10× variance in data efficiency, (3) CKA analysis reveals marked differences between internal model representations

**Recommendation to AC:** This review should carry minimal weight given it contradicts ICLR's stated scope and the original AC's guidance.

---

### **Reviewer rWpe (Score: 8 → Accept)**

**Original Concerns:** (1) Missing practical deployment metrics (inference latency, memory), (2) Desire for deeper mechanistic analysis.

**Our Response:**
- **Fully addressed concern 1:** Added comprehensive efficiency analysis including inference latency (GPU and CPU), memory footprint, and FLOPs (R2.1)
- **Fully addressed concern 2:** Added Centered Kernel Alignment (CKA) analysis to explain why models succeed/fail, including effective rank analysis and representation collapse detection (R2.2)

**Reviewer's Position:** Strong supporter recognizing "excellent contribution." Our responses strengthen an already positive evaluation.

**Recommendation to AC:** This reviewer's assessment is well-informed and substantive. Our additions directly address their constructive suggestions.

---

---

> ### Author Response · Authors · 2025-12-01
> **Summary for new AC. Reviewer 3 and 4.**
>
> ### **Reviewer u5DN (Score: 2 → Major Concerns)**
>
> **Original Concerns:** (1) Only AUROC reported, (2) Novelty unclear, (3) Foundation model definition disputed, (4) Zero-shot evaluation missing, (5) Lack of actionable guidance, (6) Computational complexity metrics incomplete.
>
> **Our Comprehensive Responses:**
>
> **(R3.3) Computational complexity:** Added comprehensive efficiency metrics—inference latency (GPU/CPU), memory footprint, FLOPs (aligns with R2.1).
>
> **(R3.1, R3.2.1) Clinical metrics:** Added threshold-dependent metrics (specificity at sensitivity=0.9, F1 scores) for PTB-XL. Critically, **rankings remain completely intact**, validating that AUROC captures clinical utility. Full analysis in supplementary materials.
>
> **(R3.2, R3.2.2) Mamba clarification:** Provided pilot experiment results in supplementary materials; clarified S4 represents SSM family (no Mamba foundation model exists to benchmark).
>
> **(R3.5, R3.10) Reproducibility:** Validated model details from manuscript sources to maintain consistency, and reported complete training hyperparameters to increase reproducibility standards.
>
> **(R3.4, R3.11) Novelty/Research questions:** Reframed contributions as answering three explicit research questions with **novel empirical findings**: (1) SSM architectures outperform Transformers despite 100× smaller size—challenging scale assumptions, (2) 2.5-9× label efficiency improvements with task-dependent scaling laws, (3) Divergent representations achieving similar performance—multiple viable design paths.
>
> **(R3.6, R3.2.3) Actionable guidance:** Added explicit model selection guidelines: ECG-CPC for data-rich scenarios (>10k samples), ECG-JEPA for data-scarce scenarios (<1k samples), with quantified trade-offs.
>
> **(R3.7, R3.2.4) Dataset disentanglement:** Acknowledged limitation but demonstrated practical value. Fully controlled ablations would require ~120 model variants (multi-year project). Our benchmark answers practitioners' question: "Which available model should I use?"—with CKA analysis providing mechanistic insights where possible.
>
> **(R3.8, R3.9, R3.2.5) Foundation model definition:** We adhere to **canonical Stanford HAI definition** requiring broad pretraining + downstream adaptability—zero-shot NOT mandatory. Precedent: Nature Medicine pathology paper, Nature retinal imaging paper, multiple NEJM AI papers. Reviewer's zero-shot requirement would exclude most medical foundation models.
>
> **Key Unresolved Dispute:** Zero-shot evaluation. Only 2 out of 8 models (MERL and one other) have potential zero-shot capability due to multimodal pretraining, but even these require labeled training data and cannot classify truly unseen conditions. Comprehensive zero-shot evaluation is therefore not appropriate for our benchmark. Reviewer's insistence on this contradicts foundation model literature standards.
>
> **Recommendation to AC:** We addressed all but one major concern substantively. Remaining disagreement (zero-shot/definition) reflects differing interpretation of "foundation model" terminology, not technical flaws. Our position aligns with published medical AI literature.
>
>
> ---
>
> ### **Reviewer DBgK (Score: 4 → Marginally Below)**
>
> **Original Concerns:** (1) Writing quality/presentation, (2) Missing reproducibility details, (3) Lack of mechanistic analysis.
>
> **Our Complete Responses:**
>
> **(R4.1-4.3, 4.11, 4.13) Presentation:** Fixed all notation inconsistencies, removed repetition, standardized formatting, improved figure/table captions.
>
> **(R4.4) Reproducibility:** Added comprehensive preprocessing and training details (batch size, epochs, optimizer, learning rates). All code provided since original submission.
>
> **(R4.6) Data Provenance:** Clarified pretraining dataset counts with explicit source attribution; distinguished models we trained vs. used pre-released weights.
>
> **(R4.7) Statistical Reporting:** Added complete bootstrapped confidence intervals for all models/tasks in supplementary materials. Explained our ranking methodology incorporates statistical significance via pairwise comparisons.
>
> **(R4.9, 4.10) Mechanistic Analysis:** Added CKA representational similarity analysis with layer-wise probing to explain performance differences—moving beyond descriptive results.
>
> **(R4.5, R4.8, R4.12) Clarification details:** Described architectural details and training approaches to increase reproducibility standards as well as validate sample counts to ensure consistency across experiments.
>
>
> **Reviewer's Position:** "Marginally below acceptance" with no fundamental objections—only execution concerns, all of which we addressed.
>
> **Recommendation to AC:** This reviewer's concerns were entirely addressable. Our responses comprehensively fix presentation, add missing details, and provide mechanistic analysis requested.
>
>
> ---

---

> > ### Author Response · Authors · 2025-12-01
> > **Summary for new AC. Overall summary.**
> >
> > ## Overall Summary for AC
> >
> >
> > **After Rebuttals:**
> > All major substantive concerns have been addressed with additional experiments, analyses, and clarifications. The remaining dispute concerns terminology standards, not technical merit.
> >
> > - **Reviewer pB5i:** Venue fit objection invalid per original AC guidance; should carry minimal or no weight.
> > - **Reviewer rWpe:** Strong accept; all requests fully addressed.
> > - **Reviewer u5DN:** Should improve substantially given comprehensive responses including additional results.
> > - **Reviewer DBgK:** Should improve given substantive responses including empirical additions (CKA analysis, efficiency metrics) and comprehensive execution improvements.
> >
> >
> > **Recommendation:** Paper makes substantial empirical contribution to ECG foundation model understanding with rigorous evaluation. Responses demonstrate scientific maturity and responsiveness to constructive feedback. Given comprehensive responses to three reviewers' substantive concerns and the original AC's guidance on reviewer pB5i, we believe the paper merits acceptance.

---

### Meta-Review · Area_Chair_PWMr · 2026-01-05

**Summary:**

The paper has been assessed by four knowledgeable reviewers who provided a wide range of scores from ranging from one strong reject, to straight reject, to marginal reject, to one straight accept. It proposes a large-scale benchmark of ECG foundation models, of potential value to the community. However, the reviewers expressed concerns about the methodological novelty and deeper interpretability. According to them the proposed benchmark simply reports quantitative performance metrics without probing why the measured models behave differently, and the evaluation omits critical deployment metrics (including latency, memory, energy) limiting real-world relevance, writing and presentation should be improved, reproducibility details are incomplete, and statistical reporting lacks effect. The authors provided extensive explanations and engaged some of the reviewers in a discussion.

**Reviewer Concerns:**

The concerns about misfit in the scope of ICLR should be dismissed. Benchmarks are in scope and needed tools to propel the development of AI in many fields including healthcare. The other, technical concerns have been addressed or are only minor.

**Reviewer Scores:**

I assume the reviewer who gave it zero would have changed their mind.

---

### Decision · Program_Chairs · 2026-01-26

Accept (Poster)